

# Event detection in finance using hierarchical clustering algorithms on news and tweets

Salvatore Carta[1], Sergio Consoli[2], Luca Piras[1], Alessandro Sebastian Podda[1] and Diego Reforgiato Recupero[1]

[1] Department of Mathematics and Computer Science, University of Cagliari, Cagliari, Italy
[2] European Commission, Joint Research Centre (DG-JRC), Ispra, Varese, Italy

## ABSTRACT

In the current age of overwhelming information and massive production of textual data on the Web, Event Detection has become an increasingly important task in various application domains. Several research branches have been developed to tackle the problem from different perspectives, including Natural Language Processing and Big Data analysis, with the goal of providing valuable resources to support decision-making in a wide variety of fields. In this paper, we propose a real-time domain-specific clustering-based event-detection approach that integrates textual information coming, on one hand, from traditional newswires and, on the other hand, from microblogging platforms. The goal of the implemented pipeline is twofold: (i) providing insights to the user about the relevant events that are reported in the press on a daily basis; (ii) alerting the user about potentially important and impactful events, referred to as hot events, for some specific tasks or domains of interest. The algorithm identifies clusters of related news stories published by globally renowned press sources, which guarantee authoritative, noise-free information about current affairs; subsequently, the content extracted from microblogs is associated to the clusters in order to gain an assessment of the relevance of the event in the public opinion. To identify the events of a day d we create the lexicon by looking at news articles and stock data of previous days up to $d^{-1}$ Although the approach can be extended to a variety of domains (e.g. politics, economy, sports), we hereby present a specific implementation in the financial sector. We validated our solution through a qualitative and quantitative evaluation, performed on the Dow Jones' *Data, News and Analytics* dataset, on a stream of messages extracted from the microblogging platform Stocktwits, and on the *Standard & Poor's 500* index time-series. The experiments demonstrate the effectiveness of our proposal in extracting meaningful information from real-world events and in spotting *hot* events in the financial sphere. An added value of the evaluation is given by the visual inspection of a selected number of significant real-world events, starting from the Brexit Referendum and reaching until the recent outbreak of the Covid-19 pandemic in early 2020.

Corresponding author
Sergio Consoli,
sergio.consoli@ec.europa.eu

# INTRODUCTION

The outbreak and the rapid growth of modern digital sources for daily current affairs, such as online newswires, microblogging websites and social media platforms, have led to an overwhelming amount of information produced every day on the Web. For this reason, Event Detection has become increasingly important in the last two decades, since it allows users to disentangle this mass of scattered and, oftentimes, heterogeneous data, and to become aware of relevant world-wide facts in a more efficient way.

From a general perspective, an event, which is somewhere referred to as *topic* (*Kumaran & Allan, 2004*), can be defined as something that happens at a particular time and place (*Allan, Papka & Lavrenko, 1998*), causing a change in the volume of textual content that discusses the associated topic at a specific time (*Dou et al., 2012*). In this sense, Event Detection aims to discover contents published on the Web that report on the same current topic, organize them in meaningful groups and provide insights, based on properties extracted automatically from the data (*Allan, Papka & Lavrenko, 1998*, *Hu et al., 2017*). It represents a valuable resource to create awareness and support decision making in various domains of application, including epidemics (*Aramaki, Maskawa & Morita, 2011*; *Rosa et al., 2020*), earthquakes (*Sakaki, Okazaki & Matsuo, 2010*), social events (*Petkos, Papadopoulos & Kompatsiaris, 2012*) and economy (see "Event Detection in Finance"), among others. In some cases, the scope of the event detection task is not limited to arranging the contents and providing analytics, but constitutes the basis for further algorithmic processing, like for example the development of automatic trading strategies in financial applications (*Gilbert & Karahalios, 2010*; *Ruiz et al., 2012*; *Makrehchi, Shah & Liao, 2013*).

Given the importance of Event Detection, an increasing number of researchers have focused their attention on this problem since the late 1990s, building on the theoretic foundations of Information Retrieval and, later on, taking advantage of the discoveries of Natural Language Processing, Text Mining and Big Data processing. Early works mainly based their approaches on traditional news stories as they started being digitalized (*Allan, Papka & Lavrenko, 1998*, *Lam et al., 2001*; *Kumaran & Allan, 2004*), while social media platforms like Twitter (http://www.twitter.com) and Stocktwits (http://www.stocktwits.com) have become the dominant data source in the last decade (*Hasan, Orgun & Schwitter, 2018*; *Atefeh & Khreich, 2015*). However, it has been demonstrated by *Petrovic et al. (2013)* that Twitter still cannot replace traditional newswire providers when considering the coverage and the timeliness of breaking news. In fact, this study shows that, while Twitter has a better coverage of minor events ignored by other media, traditional newswire sources often report events before users on the social network. Another disadvantage of microblogs is that they contain a considerable amount of noise, such as irregular syntax, misspellings and non-standard use of the language, let alone the increasing phenomenon of *fake news*, which makes it difficult to extract valuable information (*Kaufmann & Kalita, 2010*; *Ajao, Bhowmik & Zargari, 2018*). In light of this, a promising line of research has provided evidence that combining multiple sources of information allows to mitigate the flaws and exploit the advantages of each medium, thus improving the quality of the

event detection task (*Musaev, Wang & Pu, 2014*; *Petkos, Papadopoulos & Kompatsiaris, 2012*; *Thapen, Simmie & Hankin, 2016*).

Inspired by these findings, we developed a domain-specific clustering-based event-detection method that exploits the integration of traditional news articles and Stocktwits messages (which from now on will be referred to as *tweets*, for simplicity) to identify real-world events and to generate alerts for highly relevant events on a daily basis. The main intuition behind the integration of traditional press and social media is that, even though the former represents an authoritative and noise-free source which is convenient to mine to get qualitative information, it fails, taken alone, to provide insights about the entity or the resonance of the events. On the contrary, microblogs contain a considerable amount of noisy and unreliable content, but have the advantage of reflecting the impact that events have on public opinion. Because of this, we decided to exploit traditional news articles to construct a qualitative basis for our event-detection approach and to integrate the social media data on top of that, in order to get a quantitative measure.

The proposed approach, which will be described in full detail in "Proposed Approach", is defined as domain-specific because it collects news from the same sphere of interest (e.g. economy, politics, sports) and represents these documents focusing on the words that are most relevant for that field. However, the approach can be applied to various domains with minimum modifications. For example, if we are interested in identifying events that brought happiness or sadness to people, one might use social media text elements instead of news and a sentiment index indicator created on the same interval time of the social text to associate each social post to its sentiment level. Thus the lexicon would consist of tokens used within social media posts and weighted depending on their sentiment indicators. Besides, please note that our approach performs real-time event detection as it is supposed to identify events of the day $d$ without any knowledge of the future. In particular, it creates the lexicon by looking at news articles and stock data of previous days up to $d - 1$ without looking at the future. In this paper, we present the implementation of the pipeline that we designed specifically for the financial domain, which is a field where Event Detection has had one of its most promising applications (see "Event Detection in Finance"). Our motivation derives from the intuition offered by several works in the financial literature that, drawing inspiration from the *Adaptive Market Hypothesis* (*Lo, 2004*), show that public news have an impact on the stock markets, explaining a part of the return variance (*Boudoukh et al., 2019*). This justifies the need for automatic tools that can support companies, traders and all the other actors involved in the market, providing an at-a-glance visualization of acquisitions, stock splits, dividend announcements and other relevant economic events (*Hogenboom et al., 2013*).

We validated our approach through an experimental evaluation based, on one hand, on the Dow Jones' *Data, News and Analytics* dataset (https://developer.dowjones.com/site/global/home/index.gsp), which contains news articles delivered by globally renown sources, and, on the other hand, on a set of messages collected from Stocktwits, a microblogging platform inspired by Twitter, where users posts short messages related to stock markets and trading. The events that constitute our ground truth for the alert generation algorithm were selected based on the stock price time series of the *Standard &*

*Poor's 500* Index (S&P 500), following the intuition that relevant economic events lead to significant movements of the market. Our qualitative and quantitative analysis shows that the proposed method is able to extract meaningful, separable clusters, which correspond to real-world events. Furthermore, the alert generation algorithm detects *hot* events with high accuracy, proving the effectiveness of the integration of news articles and tweets.

The contributions of our work can be summarized as follows:

- we propose a novel approach to represent news documents exploiting a domain-specific lexicon created *ad-hoc* using the technique we have introduced in *Carta et al. (2020)*, where the lexicon has been generated on a different dataset;
- we design an original clustering-based event-detection approach that integrates news documents and tweets;
- we show the effectiveness of our method by means of an experimental evaluation performed on real-world datasets;
- we offer a visual inspection of the output obtained on a selected number of real-world events, including the Brexit Referendum, the U.S.-China trade war and the recent outbreak of the Covid-19 pandemic.

The reminder of this paper is organized as follows. "Related Work" offers a thorough overview of the background research on Event Detection, analyzing works that deal with different kinds of media and application fields. The proposed approach is described in full detail in "Proposed Approach". The datasets and the methodology we have carried out for the evaluation are described in "Experimental Settings" while the obtained results are illustrated in "Results". Finally, "Conclusions and Future Work" contains general conclusions about this work and future lines of research.

## RELATED WORK

The origins of Event Detection can be traced back to 1998, when a joint effort between the Defense Advanced Research Projects Agency (DARPA), the University of Massachusetts, Carnegie Mellon University and Dragon Systems aimed to define the problem within the wider field of Topic Detection and Tracking (TDT) and proposed an approach based on broadcast news stories that paved the way for new research on the field (*Allan, Papka & Lavrenko, 1998*; *Allan et al., 1998*; *Yand, Pierce & Carbonell, 1998*). Since then, a considerable variety of algorithms have been proposed to tackle the problem, gradually taking advantage of the remarkable advances in Text Mining and Natural Language Processing. Most interestingly, the birth of social media platforms like Facebook, Twitter and Stocktwits in mid 2000s and their increasing popularity, together with the birth of the new era of Big Data (*Marx, 2013*), led to a widening of the range of data that could be exploited to detect real-world events. To note that it is common to employ lexicons for news representation for the financial domain. Within our previous work (*Carta et al., 2020*), we defined a strategy to generate industry-specific lexicons from news documents with the goal of dynamically capturing the correlation between words and

stock price fluctuations. This has been then employed to solve a binary classification task with the goal of predicting the magnitude of future price changes for individual companies. Conversely, in this work we leverage the same technique to represent a new dataset and to solve a different problem, event detection.

In the following, we will hereby illustrate the previous research carried out in Event Detection, grouping it according to the type of source employed in the analysis - basically newswires, social media and an integration of heterogeneous data. Because the approach presented in this paper can be applied to different domains, also our overview of related works will cover a variety of fields, including health, security, sports and many others. However, we will conclude the overview by focusing on the financial sphere, since this is the specific domain within which our approach was developed and validated.

## Newswires-based

The first type of data that has been explored in this field consists of traditional newswires and press releases, which, however, still have a primary role even in present research. Early works typically rely on *tf-idf* features to represent the documents in a Vector Space (*Salton, Wong & Yang, 1975*; *Li et al., 2005*) or Bag-of-Words (*Zhang, Jin & Zhou, 2010*). Modification of these classic methods were proposed in order to enhance the representation by means of contextual information (*Lam et al., 2001*), lexical features (*Stokes & Carthy, 2001*), named entities (*Kumaran & Allan, 2004*), topic models (*Yang et al., 2002*) and, in more recent work, word-embeddings (*Hu et al., 2017*; *Kusner et al., 2015*). The most common approaches for the detection task are based on clustering, text classification or a combination of these (*Atefeh & Khreich, 2015*).

Going into more detail, authors in *Hu et al. (2017)* exploit word-embeddings to overcome the downsides of *tf-idf* representation, namely sparsity and high dimensionality. On top of this, they build an adaptive online clustering algorithm that leads to an improvement in both efficiency and accuracy. Similarly, authors in *Zhou et al. (2018)* enhance the *tf-idf* model by integrating the Jaccard Similarity coefficient, word-embeddings and temporal aspects of published news, with the goal of spotting *hot* events. Others (*Mele, Bahrainian & Crestani, 2019*) propose an algorithm to detect, track and predict events from multiple news streams, taking into account the publishing patterns of different sources and their timeliness in reporting the breaking news. They use a Hidden Markov Model (*Beal, Ghahramani & Rasmussen, 2002*) to represent current events and, subsequently, to predict facts that will be popular in the next time slice.

The main contribution of the proposed approach with respect to this line of research is the fact that, in our algorithm, the representation of the events extracted from news articles is enriched by the information mined on social media sources. In this way, we obtain a multifaceted perspective of events. Furthermore, another innovation regards the method employed to represent the textual data. In particular, our pipeline includes the creation of an *ad-hoc* lexical resource, which detects the words that are most relevant for a specific domain. During the construction of the vector representation of documents, only the word-embeddings of the selected terms are included, as described in full detail in "Lexicon Generation" and "Feature Engineering".

## Social media-based

Since the development of social media platforms and microblogging websites, a big share of the researchers' interest has been aimed at mining these sources of information for a more dynamic and multifaceted inspection of events. Among these platforms, the case of Twitter definitely stands out, becoming a de facto standard domain for Event Detection (*Petrovic et al., 2013*; *Saeed et al., 2019*). A thorough survey by *Hasan, Orgun & Schwitter (2018)*, focused on Twitter-based approaches, suggests that this research branch can be split into three main categories: (i) methods that exploit properties in a tweet's keywords; (ii) methods that rely on probabilistic topic models; (iii) clustering-based methods.

For the first group, it is worth mentioning TwitInfo (*Marcus et al., 2011*), TwitterMonitor (*Mathioudakis & Koudas, 2010*) and EnBlogue (*Alvanaki et al., 2011*), which identify real-time trends on Twitter and allow the final user to browse large collections of messages, providing contextual information about tweets, visualizations and meaningful insights that describe the identified topics. *Stilo & Velardi (2016)* include temporal factors in their analysis in order to cope with the limited context of Twitter messages. *Weng & Lee (2011)* propose an approach that builds signals for individual words by applying wavelet analysis (*Kaiser, 2010*) on the frequency-based raw signals of the words; this method is able to spot the most relevant words and finally cluster them to form events.

Among the works that employ probabilistic topic models to represent tweets in a latent space, TwiCal (*Ritter, Etzioni & Clark, 2012*) is an open-domain event-extraction framework that identifies significant events based on a multitude of features including, but not limited to, contextual, dictionary and orthographic features. TopicSketch (*Xie et al., 2016*) is a system that identifies bursty topics from live tweet streams in an efficient way, by tracking the occurrence of word pairs and triples in small "sketches" of data. *Zhou, Chen & He (2015)* devise a lexicon-based approach to spot tweets that are event-related and, based on these tweets, extract a structured representation of events by means of an unsupervised Bayesian model.

As for clustering-based approaches, *Petrović, Osborne & Lavrenko (2010)* propose a time-efficient way to determine the novelty of a new tweet appearing in a live stream; novel tweets represent new stories and, therefore, will be assigned to newly created clusters, which are later ranked according to the number of unique user posts and the entropy information. The approach by *Becker, Naaman & Gravano (2011)* groups tweets into semantically related clusters and then exploits a series of cluster properties (including temporal, social and topical features) in order to discriminate between real-world events and non-events messages. Analogously, *Kaleel & Abhari (2015)* employ a locality-sensitive-hashing scheme to extract clusters from the Twitter stream; the exploration of the clusters, which takes into account size, time and geolocation, leads to the identification of significant real-world events.

As already mentioned, the novelty of our approach with respect to these works is that social media data is not considered on its own, but in conjunction with news articles, in order to obtain a more insightful representation of events.

## Integration of heterogeneous data

As stated in the Introduction section, several works in the literature suggest that, in many scenarios, an integration of different kinds of sources is necessary to improve the effectiveness of the event-detection algorithm, as far as both timeliness and coverage are concerned (*Petrovic et al., 2013*; *Musaev, Wang & Pu, 2014*; *Petkos, Papadopoulos & Kompatsiaris, 2012*). As a consequence, a promising research branch has grown based on this principle. One interesting example is represented by the work by *Osborne et al. (2012)*, which aims to mitigate the spuriousness intrinsic to Twitter messages by means of information from Wikipedia. The latter is used as a filter to discard large numbers of noisy tweets, thus refining the representation of the extracted events. *Thapen, Simmie & Hankin (2016)* propose a methodology to automatically spot outbreaks of illness from spikes of activity in real-time Twitter streams. A summary of these events is provided to the user with the goal of creating situational awareness; this is achieved by presenting the most significant tweets and by linking them with relevant news, which are searched on the Web based on term occurrences. *Petkos, Papadopoulos & Kompatsiaris (2012)* develop a novel multimodal clustering algorithm to explore multimedia items extracted from several social media platforms, with the purpose of detecting social events. The authors suggest that the proposed approach can be extended to any scenario which requires the usage of multimodal data. In (*Consoli et al., 2010*; *Consoli et al., 2020*) the authors present some novel optimization strategies for the quartet method of hierarchical clustering, a methodology popular in the context of biological phylogenesis construction by integration and clustering of different heterogeneous data.

Our approach differs from other works in this category in the way news and tweets are juxtaposed. In fact, the information extracted from news articles constitutes the basis of our event-detection algorithm, while the processing of tweets is implemented on top of that, with the goal of corroborating that information.

## Event detection in finance

Event detection, Natural Language Processing and Sentiment Analysis have been widely applied in the financial sphere to provide more and more insightful tools for supporting decision making (*Xing, Cambria & Welsch, 2018*). Some works have pushed the research as far as correlating the information about the events with the movement of the stock prices, with the goal of predicting future returns and developing trading strategies. *Heston & Sinha's (2017)* study in which way the sentiment and the aggregation of the news affect the time horizon of the stock return predictability. In particular, through a neural network-based method, they show that daily news can forecast returns within one or two days, while aggregating news over one week provides predictability for up to 13 weeks. Moreover, the authors produce evidence that positive news stories increase stock returns quickly, while negative stories have a long delayed reaction. *Schumaker & Chen (2009)* combine news textual data and S&P 500 price time-series to estimate a discrete stock price twenty minutes after a news article was released, using Support Vector Machines (*Suykens & Vandewalle, 1999*). *Ding et al. (2015)* extract a structured representation of events from financial news, relying on the Open Information Extraction

tool developed by *Yates et al. (2007)*, and subsequently train a neural tensor network to learn event embeddings; this dense vector representation is then fed into a deep learning model to predict short-term and long-term stock price movements on S&P 500.

As far as social media-based approaches are concerned, *Daniel, Neves & Horta (2017)* carry out an analysis of the content published on Twitter about the thirty companies that compose the Dow Jones Average. In particular, the authors start by detecting and discarding noisy tweets that might distort the information about relevant financial events; in the next steps, they perform a sentiment analysis on the valuable tweets and correlate them with the behavior of the stock market. *Tsapeli et al. (2017)* apply a bursty topic detection method on a stream of tweets related to finance or politics and, then, employ a classifier to identify significant events that influence the volatility of Greek and Spanish stock markets. Events are represented as feature vectors that encompass a rich variety of information, including their semantics and meta data. Starting from the same motivations, *Makrehchi, Shah & Liao (2013)* collect a set of tweets related to companies of the S&P 500 index and label them based on the price movement of the corresponding stock. Then, they train a model on this set to make predictions on the labels of future tweets and, on top, create trading strategies that prove to give significant returns compared to baseline methods.

Another branch in financial event detection is focused on the extraction of potentially useful information, like events, from news and social media, that can represent a valuable resource for further algorithmic processing or for human-in-the-loop decision making. The Semantics-Based Pipeline for Economic Event Detection (SPEED) (*Hogenboom et al., 2013*) aims to extract financial events from news articles and annotate them with meta-data with an efficiency that allows real-time use. This is achieved through the integration of several resources, including ontologies, named entities and word disambiguators, and constitute a feedback loop which fosters future reuse of acquired knowledge in the event detection process. *Jacobs, Lefever & Hoste (2018)* tackle the task of economic event detection by means of a supervised data-driven approach. They define the problem as a sentence level multilabel classification task, where the goal is to automatically assign the presence of a set of pre-determined economic event categories in a sentence of a news article. Following the same intuition, *Ein-Dor et al. (2019)* develop a supervised learning approach for identifying events related to a given company. For this purpose, the authors train a sentence-level classifier, which leverages labels automatically extracted from relevant Wikipedia sections.

*Hogenboom et al. (2015)* measured the effects of various news events on stock prices. They retrieved 2010 and 2011 ticker data and news events for different equities and identified the irregular events. Finally, they cleaned the ticker data of rare event-generated noise and obtained a dataset with a more accurate representation of the expected returns distribution.

Moreover, *Nuij et al. (2014)* presented a framework for automatic exploitation of news in stock trading strategies where events were extracted from news messages presented in free text without annotations. It turned out that the news variable was often included in the optimal trading rules, indicating the added value of news for predictive purposes.

The innovation that we bring with respect to the literature consists, first of all, in the integration of different sources to obtain richer representations of events. Second, we propose a method to estimate the resonance of an event based on the activity on social media platforms, and we leverage this measure to provide warnings to the final user. Last but not least, our method has been deployed for real-time detection of financial events, accordingly within the evaluation we carried out we applied it on historical data but without considering information beyond the day under analysis.

## PROPOSED APPROACH

The problem that we set out to tackle in this work is twofold. In the first place, we want to identify groups of news stories related to real-world events of a specific domain, on a daily basis. More precisely, given a day $d$ and a look-back period of $n$ days, our approach aims to extract $k$ semantically related clusters made of text documents published by newswires providers during the $n$ days before $d$. The parameter $k$ is automatically estimated from the data so that it reflects the real events actually taking place in the best possible way. Each cluster is described by a set of properties, including relevant headlines and keywords, that are semantically correlated with the event represented by the cluster.

Secondly, we intend to tackle the problem of understanding whether a highly relevant event is taking place on a given day. Such an event is defined as *hot* and is associated with an increased amount of content published on a microblogging platform about that topic in the same time interval.

The main ideas underlying our proposed approach are the following:

- detecting the words that are more significant for the context under analysis can lead to more effective domain-aware representations of documents;
- clustering techniques allow to identify and distinguish events reported in news stories;
- the integration of social media data and news stories is key to spot *hot* events that are potentially noteworthy for the context under analysis.

In the following sections, we will describe the implementation of the algorithm that we designed for a specific scenario, namely the financial field. However, we would like to point out that our proposal can be generalized to any sphere of interest with minimum modifications, concerning mainly the filter applied to the news corpus and the numeric feedback used to assign a score to words in the lexicon generation phase.

### Overall architecture

The proposed algorithm is outlined in the pipeline in Fig. 1, which is repeated for each single day $d$ on which the event-detection task is executed. The first step consists of the generation of a dynamic, context-specific lexicon, which includes the list of words that have proven to have the biggest impact on the market in a given period before $d$ (*Carta et al., 2020*). This resource is computed by combining two different data sources: on the one hand, words are extracted from financial news published in a time interval that typically ranges from 2 to 4 weeks previous to $d$. On the other hand, the stock price time-

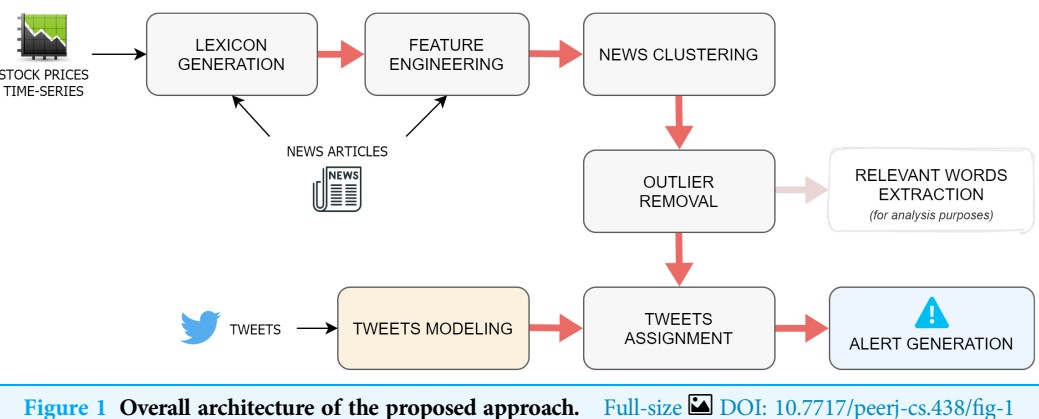

**Figure 1 Overall architecture of the proposed approach.**

series of the chosen market is used to assign numeric scores to the words appearing in the press releases.

Once the specialized lexicon is obtained, it is applied as a filter on the news documents, so that only the terms that appear in the lexicon are retained. Subsequently, a document-embedding representation of each news story is constructed by computing the average of the word-embeddings of its filtered words.

After the news-modeling stage, the document-embeddings are fed to an agglomerative clustering algorithm, which returns a list of labels, which indicate the cluster to which each specific observation belongs, and a variable number of centroids. Intuitively, each cluster should correspond to the event discussed in the news contained in it, while the cluster centroid serves as an high-level discriminating representation of the event. The previous output is later refined through an operation of outlier removal, whose goal is to find and discard those documents whose assignment to their cluster is weak. Once the spurious data have been cleaned out from the clusters, a series of properties are extracted from each group of news, both for illustrative and for evaluation purposes. This information includes the titles of the articles, the percentage of positive and negative words (associated to high or low stock price variations, as described in the next paragraph), and the list of the most relevant words for the cluster, assessed through a *tf-idf*-based method.

At this point of the pipeline, the integration between news stories and social media data takes place. The idea here is, first, to find tweets that are semantically correlated to some group of news and, second, to detect if an event reported in the news has a wide resonance on the social media platform. More specifically, the tweets relevant for the market under analysis published on the most recent day of the time interval are collected and then represented with the same embedding-based method previously employed for the news. The assignment task consists of attaching every tweet to the closest news-cluster, according to a similarity measure calculated between the tweet-embedding and each news-centroid, as long as this distance is smaller than a defined *tweet distance threshold*; otherwise, the tweet is discarded.

The last step in the event-detection pipeline is the alert generation. This happens when the percentage of the assigned tweets w.r.t the overall number of tweets published on the

most recent day of the time interval is bigger than a given *alert threshold*. In fact, this suggests that a considerable number of people on the social media platform are discussing some events reported in the news.

## Lexicon generation

The lexicon generation stage leverages the method that we proposed in *Carta et al. (2020)*, which we hereby set out to illustrate for the sake of completeness. From a general perspective, the goal of the lexicon generation is to select the set of words that are most relevant for a specific domain in a given time interval. In order to be able to capture the impact of events that occur day by day (and thus the effect of new words that show up in news articles reporting such events), we perform the lexicon creation in a dynamic way, repeating its generation every day. For these reasons, we define the lexicons generated by our approach as *time-aware* and *domain-specific*.

If we apply this concept to the financial sphere, the relevance of a word can be estimated by observing the effect that it has on the market after the delivery of the news stories containing this word. In this sense, the resulting lexicons will capture potential correlations between words that appear in news stories and stock price movements: terms that are consistently followed by significant positive (negative) variations will receive a high (low) score, while terms that are followed by negligible or arbitrary variations will tend to have a score close to 0. Going into more detail, for each day we collect all the news that are relevant for the S&P 500 Index published during the time frame $[d - l, d - 1]$ (with $l \geq 1$). More precisely, we select all news with at least one mention of *Standard & Poor* (or strictly related keywords like *SP500* and *SPX*). For each news article in this set, we extracted the text, consisting of the title, the snippet and the full body of the article, and then we performed some standard pre-processing techniques on it, such as stop-words removal (using that of Stanford CoreNLP (https://tinyurl.com/yygyo6wk)), stemming and tokenization (the last two using NLTK (https://www.nltk.org/)). In addition, we removed from the corpus all the words that appeared too frequently and too infrequently, according to given tolerance thresholds. In our case, we filtered out all the words that appear in more than 90% of the documents or in less than 10 documents (both thresholds were set experimentally). Subsequently, we construct a document-term matrix, in which each row corresponds to a news article and date and each column corresponds to a term, as obtained after the pre-processing. In the next step, we iterate over the rows of the matrix and, for each of them, we assign to each of its terms a value equal to the stock price variation registered on the day after the article was published, defined as:

$$\Delta_{d'} = \frac{close_{d'} - close_{(d'-1)}}{close_{(d'-1)}}, \tag{1}$$

where $d' \in [d - l, d - 1]$ is the day after the publication of the underlying article, and $close_{d'}$ is the price of the stock at the closing time of the market on day $d'$. Finally, each column is averaged (counting only non-zero entries), thus obtaining a list of terms, each associated to a score given by the average of the values assigned to them. We sort the terms by decreasing scores and select the first $n$ and the last $n$. These are the ones associated to

higher price variations, respectively positive and negative, and represent the time-aware, domain-specific lexicon that will be exploited for the news modeling phase.

Hereafter, we give some formal notation to illustrate how this step corresponds to perform a marginal screening (*Genovese et al., 2012*), a form of variable selection which is proven to be more efficient than the Lasso and with good statistical accuracy. Let us assume that in the period $[d - l, d - 1]$ the algorithm collects $N$ articles, where a portion of them contains the term $j$. Then

$$f(j) = \frac{1}{N} \cdot \sum_{1 \leq k \leq N} X_k(j) \cdot \Delta_d(k),$$

where $X_k(j)$ is a dummy variable for whether term $j$ appears in article $k$ and $\delta_d(k)$ is the return on the day $d$ for article $k$. In this form, $f(j)$ is the slope of a cross-article regression of $\delta_d = (\delta_d(1),\ldots,\delta_d(N))$ on the dummy variable $X(j) = (X_1(j),\ldots,X_N(j))$. More precisely, $f(j)$ are coefficients of a marginal regression. By sorting them by decreasing scores and selecting those whose values are over (under) some specified threshold $t^+$ ($t^-$), is similar to taking the first $n$ and the last $n$. Moreover, in our lexicon construction, if $S$ is the index set of positive and negative words (those corresponding to high or low stock variations), and $\hat{S} = \{j : f(j) \geq t^+ \text{ or } f(j) \leq t^-\}$, under certain conditions $Prob(\hat{S} = S) = 1$ as $N$ and the number of terms go to infinity. This corresponds to the sure screening property (*Fan & Lv, 2008*).

### Feature engineering

The aim of the news modeling phase is to obtain a representation of the news documents in a vector space, such that it captures its semantics and it is convenient for the subsequent cluster analysis. This must be achieved by characterizing each article through the words that are more relevant for the specific domain, ignoring the words that represent noise or that, in any case, do not provide valuable information. The two main resources that are exploited in this stage are the lexicons described in the previous "Lexicon Generation" and a word-embedding model, which assigns a dense vector representation to words (*Mikolov et al., 2013*). The latter can be obtained by training the model on the text corpus under analysis or by loading a pre-trained model previously fit on an independent corpus.

First of all, each news article undergoes a series of standard text pre-processing operations, namely tokenization, conversion to lower case and stopwords removal. Subsequently, the words of each document are filtered by means of the lexicon produced on the day of the publication of the news, so that only the words that appear in the lexicon are retained. Finally, the word-embeddings of the filtered words are extracted and their average is computed to obtain the news-embedding.

### News clustering

The embedding representation of news documents obtained in the previous step is the input to the clustering algorithm (Fig. 2), whose goal is to split the articles in semantically-correlated groups. Ideally, each cluster corresponds to a real-word event.

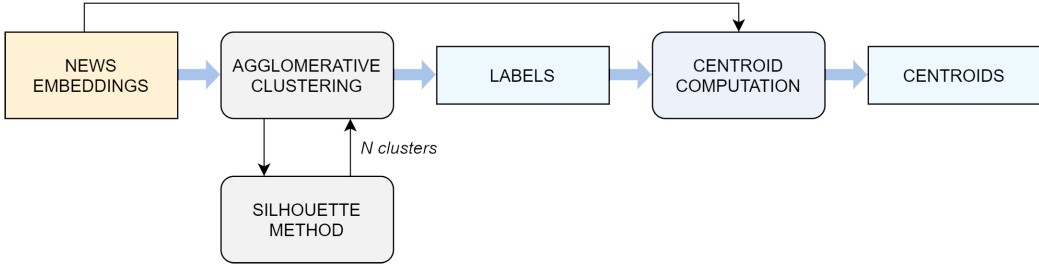

**Figure 2  Flowchart of the clustering algorithm.**   

For this purpose, we employ the *agglomerative clustering* algorithm. The decision mainly arises from a comparison with other standard techniques, which in this specific scenario do not prove as effective at separating the input data (see "Experimental Settings" for a detailed comparative analysis). The agglomerative clustering is a method pertaining to the family of hierarchical algorithms, which build nested clusters by merging or splitting them successively (*Rokach & Maimon, 2005*; *Murtagh, 1983*; *Zhao, Karypis & Fayyad, 2005*). More specifically, the agglomerative algorithm follows a bottom up approach: at the beginning, each sample represents a cluster on its own, and clusters are successively merged together according to a linkage criteria. In this study, the choice fell on the average linkage criterion, which minimizes the average of the distances between all observations of pairs of clusters, while the affinity used to compute the linkage was the cosine distance, the most commonly employed metric when dealing with text documents.

An important aspect to take into account is the number of clusters $k$ that the algorithm extracts. This can be set as a parameter to the agglomerative method, but finding the most suitable $k$ a priori is not trivial. Above all, using a fixed $k$ for all days would in most cases lead to a misshaped approximation of reality, because the number of events taking place around the world naturally varies enormously from day to day and in different periods of the year. For this reason, a technique known as the *silhouette maximization method* is used to find the ideal value of $k$ in a dynamic manner. The silhouette coefficient is a metric used to evaluate the performance of a clustering algorithm when a ground truth is not available. It ranges from −1 to 1, where higher scores relate to models with better defined clusters and it is defined for each sample by the following formula:

$$silhouette = \frac{(b - a)}{max(a, b)},$$

where $a$ is the mean distance between a sample and all other points in the same class and $b$ is the mean distance between a sample and all other points in the *next nearest cluster*. A global score for the whole model can be easily computed as the average of all the scores computed on the single samples. In fact, the average silhouette coefficient is the metric that guides us in the choice of the best number of clusters $k$ on each day on which the event-detection pipeline is executed. The agglomerative clustering algorithm is run with

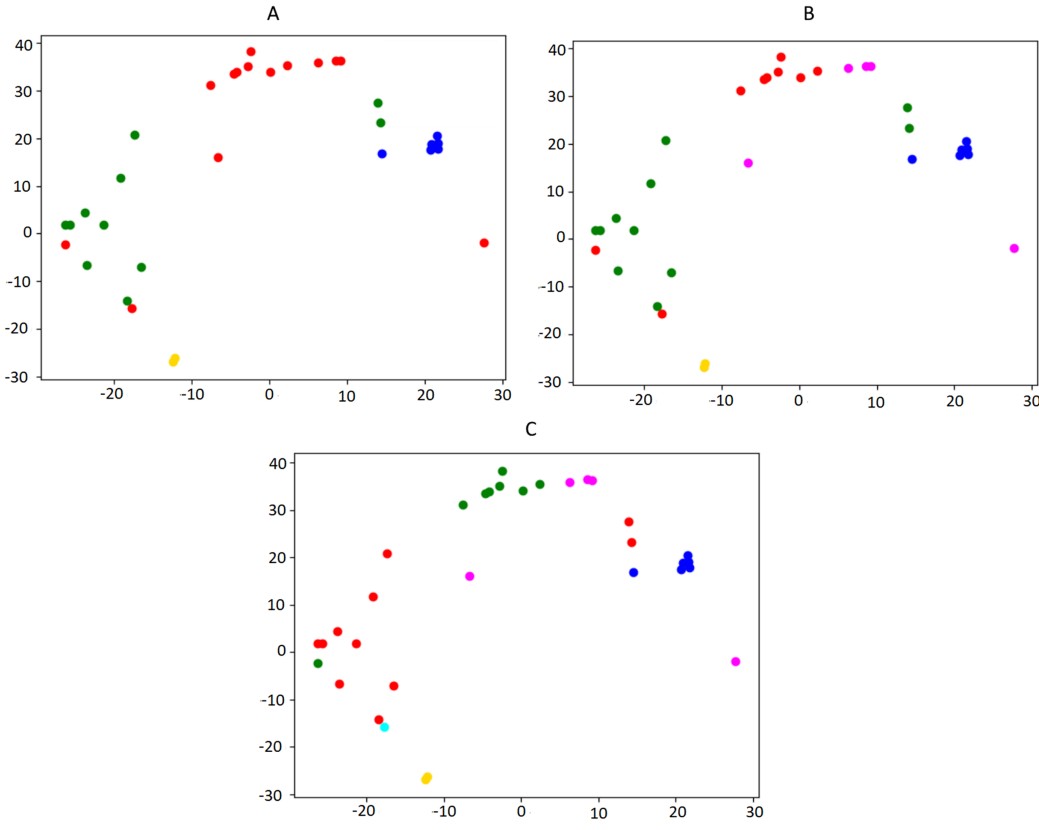

**Figure 3 Illustration of the silhouette maximization method.** For space reasons, only the output with 4, 5 and 6 clusters is showed (A, B, C, respectively). In this case, the algorithm would choose the number of clusters k = 5, which is the value that leads to the highest silhouette score (0.27 against 0.24 in the other two settings). The bi-dimensional visualization of news clusters is obtained by means of t-SNE, a tool to visualize high-dimensional data (*Van der Maaten & Hinton, 2008*), which reduces the dimension of embeddings from 300 to 2. Every point represents a news in the 2D space and each color represents a different cluster.

$k$ values ranging from 2 to 10 and the silhouette score is computed on the output for every $k$. The value of $k$ which led to the highest silhouette is selected.

Figure 3 illustrates the output of a small instance of the silhouette maximization method applied on a set of news collected in one week.

The output of the agglomerative algorithm is simply a sequence of labels, which indicate the cluster to which each specific observation belongs. The method by itself does not return any centroid, as this notion is not employed in any step of its procedure. However, the next phases in the event-detection pipeline require also a centroid for each cluster (i.e., a vector obtained through a combination of the samples in the cluster, typically the mean or median). For this reason, we manually construct a series of centroids, computed as the median of the document-embeddings contained in the respective cluster. In this scenario, the median is a more suited choice compared to the mean, because it is less sensitive to noise and outliers. The resulting centroids, which are vectors of the same length of the document-embeddings, serve as high-level discriminating representations of the corresponding events.

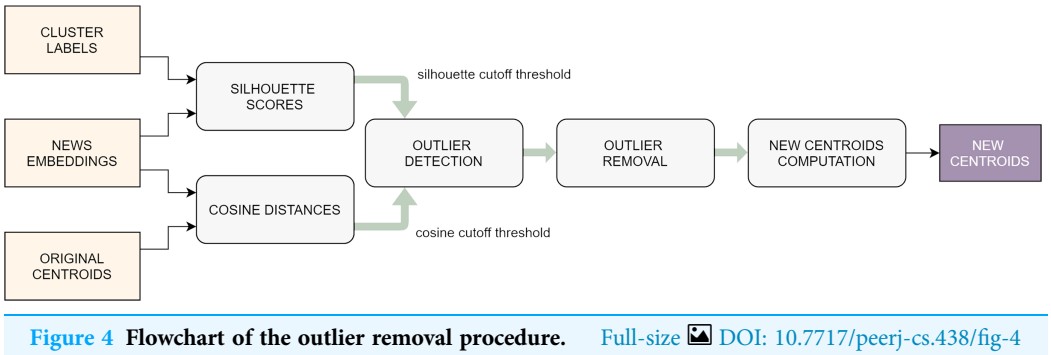

**Figure 4 Flowchart of the outlier removal procedure.**

## Outlier removal

Not necessarily all the articles published by press sources report events currently taking place: for example, in some cases they might refer to anniversaries of past happenings or they might discuss current affairs from a general perspective, including more than one event. This can cause noise in the formation of the clusters and, to some extent, can negatively influence the features of the centroid. For this reason, it is recommendable to detect and remove the outlier documents within each cluster (Fig. 4). Intuitively, these are the observations on which the clustering algorithm was least effective.

Again, the silhouette coefficient (this time in its per-sample version) is used to spot the documents that were poorly clusterized: those with lower silhouette scores are typically the ones that lie on the border between two or more groups, causing a higher uncertainty in the clustering task. This is not enough, though: in fact, there might be samples that, even if they are not located on a border, have a weak correlation with the other articles of the same cluster: these are typically the documents that lie further away from the centroid of the group to which they belong. Therefore, the noise-reduction task that we designed exploits two different metrics in order to detect the outliers: the per-sample silhouette coefficient and the cosine distance from the centroid. First of all, the samples are sorted in decreasing order according to these two metrics, respectively, thus obtaining two different rankings. Then, *cutoff threshold* is defined on each ranking, by picking a percentile value computed on each of the two lists, respectively (typically somewhere between the 10th and the 30th). Finally, all the samples whose scores are below the *cutoff threshold* in one of the two rankings are marked as outliers and suppressed. It is straightforward to note that choosing higher percentiles to set the *cutoff threshold* will make the algorithm more selective, in the sense that it will consider more documents as outliers. In rare extreme cases, this might lead to the total suppression of one of more clusters, if these already contained few samples in the first place and were not compact.

At this point, the new centroids of the affected clusters need to be computed, to account for the elimination of some of the documents. Similarly to what was done before, each centroid is obtained as the median of the document-embeddings that remain in the cluster after the outlier removal.

An example of clustering and outlier removal can be observed in Fig. 5, which presents a bi-dimensional visualization of the clusters obtained from the financial news published on

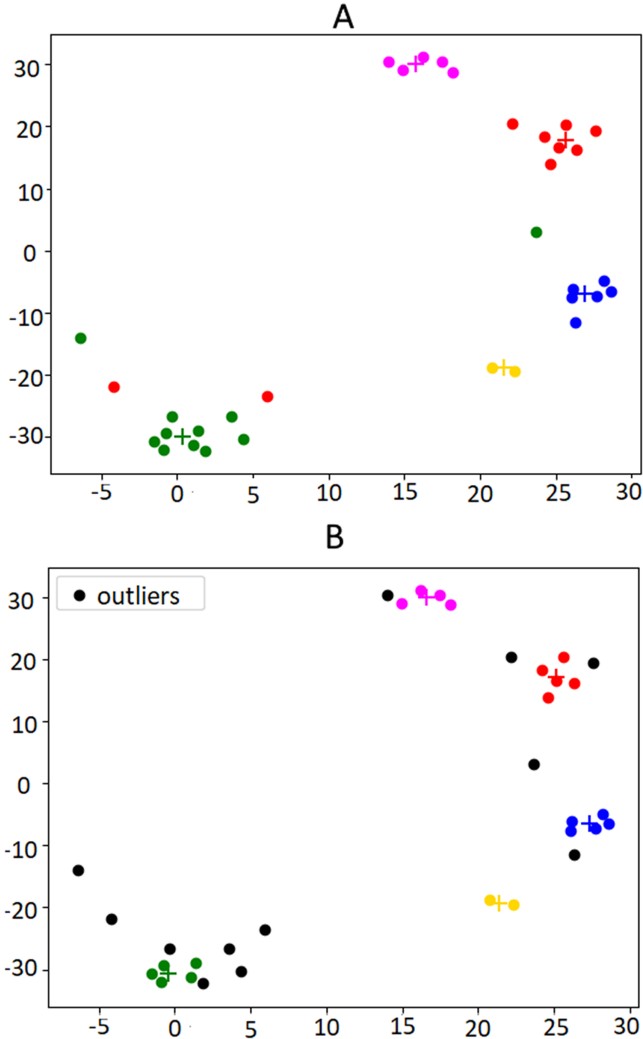

**Figure 5 Illustration of the outlier removal method on the weeks published in the week before the Brexit referendum.** (A) shows the original clusters including all documents. In (B) the outliers are marked in black. For this example, the 30th percentile was used as the cut-off threshold, in order to make the effects of the algorithm more visible. Centroids are indicated by "+" marks, in the same color of the respective cluster.               

the U.S. press on the week before the Brexit referendum, an event that we will use along the paper that took place on the 23rd of June 2016.

## Relevant words extraction

There are several properties that can be extracted from each cluster to provide insightful information to the user. At the same time, they can serve as a useful means to perform a qualitative evaluation of the clusters, as they allow judging at first sight if the clusters are meaningful and coherent. These properties include the titles and snippets of the news articles, the time-span covered by the cluster, the percentage of positive and negative words from the specialized lexicon and the list of relevant words. Hereby we focus our attention on the latter (Fig. 6), as the other ones are trivial to extract.

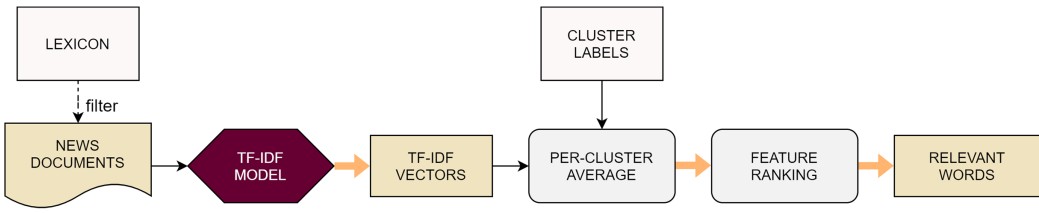

**Figure 6 Flowchart of the algorithm used to extract the most relevant words from each cluster.**

**Table 1 Lists of three most relevant titles (i.e., pertaining to the three documents that are closest to the respective centroids) for each of the five clusters obtained from the news collected in the week before the Brexit referendum (cluster#3 contains only two news documents in total).**

| Cluster | Top-3 titles |
|---|---|
| #0 | - Markets Rise on U.K. Polls—Jumpy investors shift their bets as opinion surveys tilt slightly to Britain staying in EU |
| | - Relief Rally Lifts Stocks and Oil—Dow industrials gain 129.71 as bets rise that U.K. would stay in EU; crude jumps 2.9% |
| | - Global markets rally as polls show that enthusiasm for Brexit is waning |
| #1 | - D.C. juggernaut in manufacturing is splitting in two |
| | - Global Finance: Abu Dhabi Banks Considering Merger — Deal would create biggest lender in Middle East; industry stocks rally in region |
| | - Global Finance: Bankruptcy Filing By Phone Firm Hits Big Brazilian Bank |
| #2 | - As Fears of Brexit Ease, Wall St. Thrives |
| | - Health and Tech Shares Lead a Down Day for the Market |
| | - Market Ends a Losing Streak |
| #3 | - This Time Around, the Volatility Index Matters |
| | - Stock Volatility Expected to Last |
| #4 | - Stocks Fall 5th Day in Row—Fed rate decision likely means high-dividend shares will benefit as banks are pressured |
| | - Growth Tepid, Fed Slows Plan to Raise Rates |
| | - Brexit fears lead Fed to postpone increase in key interest rate |

At first, all the news articles included in the current time interval are fed to a *tf-idf* model, regardless of their cluster. The features used to fit the model are the words included in the specialized lexicon, so this is equivalent to filtering the documents' words with the lexicon. The output of the model is a sequence of vectors, one for each document, where the values represent the relevance of the corresponding words for the document. At this point, the *tf-idf* vectors are grouped up according to the cluster to which the respective documents have been assigned. Then the average of the vectors is computed for each group, thus obtaining a unique score for each feature for each cluster, indicating the relevance of that word for that cluster. Finally, it is sufficient to rank the features and select the top *n* to get the list of the most relevant words for each group of news.

Tables 1 and 2 show the instance of the 3 most relevant headlines and the lists of the 10 most relevant words, respectively, for the clusters obtained on the day of the Brexit referendum. It is clear from the news titles that cluster#0 effectively captures this event, while the others are more focused on different aspects of business and finance (cluster#1 deals with global finance, cluster#2 with stock markets, cluster#3 with volatility, cluster#4 with the Federal Reserve system). The fact that several mentions of Brexit also appear in

**Table 2 List of the 10 most relevant words for the cluster obtained on the day of the Brexit referendum.**

**Clusters**

| #0 | #1 | #2 | #3 | #4 |
|---|---|---|---|---|
| polls | capital | cents | volatility | wednesday |
| eu | firm | gallon | options | economic |
| leave | business | yen | exchange | policy |
| stay | based | us | short | rise |
| british | protection | feet | indication | inflation |
| referenum | owns | cubic | matters | december |
| rising | majority | copper | rising | july |
| momentum | commercial | heating | historical | won |
| surveys | filing | wholesale | problem | results |
| volume | value | silver | worried | broader |

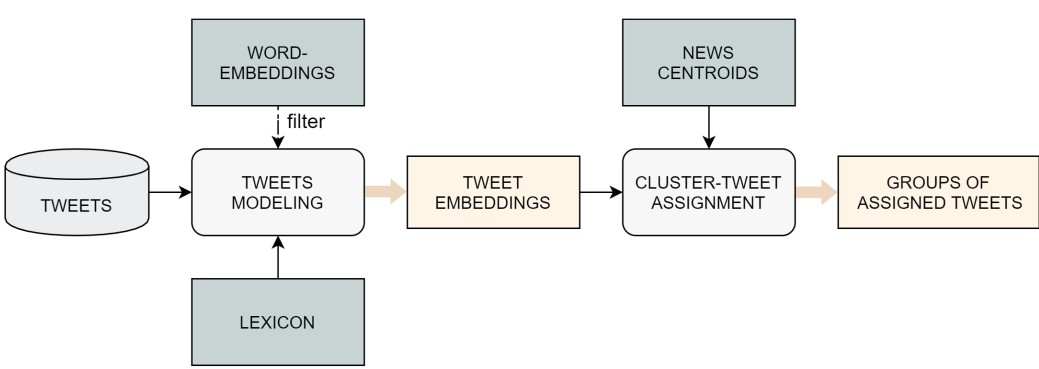

**Figure 7 Flowchart of the tweet assignment task.**

the headlines of the other clusters is attributable to the huge impact of the British referendum on many spheres of economy around the world. Not surprisingly, also the titles of the Brexit-cluster are characterized by the financial jargon, since the whole set of news on which the event-detection task is performed was selected by this specific field of interest. For the same reason, the variety of semantic fields involved in the lists of relevant words is not so wide between clusters. Noticeably though, these lists reflect quite accurately the content of headlines of Table 1.

## Tweet assignment

The goal of this phase is to enrich each cluster of news with a group of tweets that are semantically correlated with the event associated to the cluster (Fig. 7). First of all, we collect from Stocktwits all the tweets relevant to the market under analysis, published on the most recent day of the time interval used for the event-detection task[1]. The duplicate tweets are removed in order to avoid the negative influence of spam. A vector representation is constructed for every tweet with the same method used for the news

[1] This can be easily done by using the *cashtag* functionality, i.e. by searching for tweets that contain the symbol <dolloar> followed by the market code.

**Table 3** List of the 3 most relevant tweets (i.e. closest to the respective centroid) for each of the 5 clusters obtained from the news collected in the week before the Brexit referendum.

| Cluster | Top-3 Assigned Tweets |
| --- | --- |
| #0 | The polls are closer than the establishment cares to admit |
| | https://www.reuters.com/article/uk-britain-eu-tns-poll-idUKKCN0Z824K |
| | Cameron and Osborne have credibility issues with British |
| | http://www.express.co.uk/news/uk/682561/david-cameron-eu-referendum-european-union |
| | -brexit-germany-boris-johnson-brussels |
| | EU referendum outcomes explained |
| | https://www.youtube.com/watch?v=VRIF4C_c2qs |
| #1 | No tweets assigned. |
| #2 | No tweets assigned. |
| #3 | Gotta love this crazy volatility market |
| | S&P 500 squeeze back to 208 #volatility |
| | Fundamentals Still Look Solid Despite Brexit-Induced Volatility |
| #4 | No tweets assigned. |

articles: the punctuation is removed, the text is tokenized, the words are filtered with the specialized lexicons and the average of the embeddings of the remaining words is computed.

Subsequently, the actual assignment takes place. Each tweet-embedding is compared to each news-cluster centroid using the cosine similarity measure. The tweet is attached to the closest cluster only if this distance is smaller than a fixed *tweet distance threshold*; otherwise, the tweet is considered as noise and is not associated to any cluster.

An example of tweet assignment can be observed in Table 3, which presents the lists of the 3 most relevant tweets for the clusters obtained on the day of the Brexit referendum. Most importantly, the content of these tweets is totally coherent with the titles reported in Table 1. This means that the association of tweets to news-clusters was successful. It is noteworthy that even URLs, in case they contain meaningful keywords, can provide precious information for the semantic representation of the tweets and for the assignment task. This can be observed in the URLs of the first two tweets of cluster#0, which contain direct references to the Brexit referendum.

## Alert generation

The last step in the pipeline consists of the detection of the *hot* events: these are facts that not only have been reported in the news, but are also widely discussed on the social media platforms. The amount of content produced on the Internet about a certain episode is an insightful indicator of the entity of that episode and its consequences. For example, a remarkable popularity of a certain event among the users of Stocktwits is likely to translate into a potential impact on the market, since this website deals mainly with business and finance. Hence the importance of generating alerts that make the

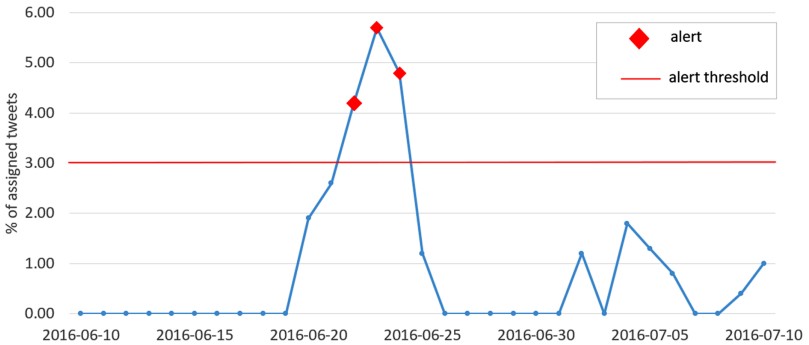

**Figure 8 Plot of the percentage of assigned tweets (among all clusters) with respect to the overall number of published tweets, for each day in the interval between the 10th of June 2016 and the 10th of July 2016.** The red markers indicate the generated alerts, while the red horizontal line represents the alert threshold.

investor or trader aware of factors that they should take into account before operating on the market.

This task exploits the tweets-cluster assignment produced in the previous step and simply checks if the percentage of assigned tweets (among all clusters) with respect to the overall number of tweets published on the most recent day of the time interval (thus including also the discarded tweets) is above a fixed *alert threshold*. If this is true, an alert is generated.

The plot in Fig. 8 shows the percentage of assigned tweets between the 10th of June 2016 and the 10th of July 2016. As expected, a peak is observed on the 23rd of June, day of the Brexit referendum, and an alert is generated.

# EXPERIMENTAL SETTINGS

In this section we will illustrate the datasets we have employed within our study and the methodology we have followed for the experimental evaluation.

## Datasets

### Dow Jones DNA

The Dow Jones "Data, News and Analytics" dataset (https://developer.dowjones.com/site/global/home/index.gsp) provides documents from more than 33,000 globally renowned newspapers, including e.g. *The Wall Street Journal*, the *Dow Jones Newswires* and *The Washington Post*. The publications are both in print and online format and cover a wide variety of topics, such as finance, business, current affairs and lifestyle. The delivery frequency ranges from ultra-low latency newswires to daily, weekly, or monthly editions. For every article in the dataset, the headline, the snippet and the full body are available. Furthermore, every item is enriched with a set of metadata providing information about the source, the time and place of the publication, the relevant companies and the topics, among others.

Content usage rights vary based on the specific content, API, or feed combination. These rights include the display for human consumption or text mining for machine consumption and the content retention period.

### Stocktwits data

Stocktwits (http://www.stocktwits.com) is a social media platform designed for sharing ideas between investors, traders, and entrepreneurs. It was founded in 2008 and currently counts over two million registered community members and millions of monthly visitors. Inspired by Twitter, it allows users to share and explore streams of short messages with a maximum 140 characters, organized around tickers referring to specific stocks and financial securities in general. This is achieved through the use of *cashtags*, which consists of the symbol "$" followed by the code of a financial security (e.g., "$AAPL", "$FB").

The dataset that we employed in our study contains the entire stream of tweets about S&P 500 published between June 2016 and March 2020. These messages were downloaded by means of the official API (api.stocktwits.com/developers/docs), selecting only the ones that contained the cashtag "$SPX", which corresponds to the aforementioned stock. The whole obtained collection contains 283,473 tweets.

Beside the full text of the tweet, every item in the dataset comes with a set of metadata, including the exact time of the publication, the number of "likes" (positive reactions by other users) received by the tweet, the sentiment score associated with the content and the number of the author's followers.

### Standard & poor's time-series

Another fundamental data source exploited in our analysis consists of the stock price *time series* of the the Standard & Poor's 500 Index, which measures the market performance of 500 large companies listed on stock exchanges in the United States. Companies are weighted in the index in proportion to their market value. The 10 largest companies in the index account for 26% of the market capitalization of the index. These are, in order of weighting, Apple Inc., Microsoft, Amazon.com, Alphabet Inc., Facebook, Johnson & Johnson, Berkshire Hathaway, Visa Inc., Procter & Gamble and JPMorgan Chase.

The dataset that we used for our evaluation was collected at a daily frequency and includes the following information:

- *open*: price of the stock at the opening time of the market;
- *close*: price of the stock at the closing time of the market;
- *high*: maximum price reached by the stock during the day;
- *low*: minimum price reached by the stock during the day;
- *volume*: number of operations performed on the stock during the day.

The aforementioned indicators are collected in an aggregated way, taking into account the values recorded for all companies included in the index.

## Methodology and settings

The goal of the experimental framework that we designed to evaluate the proposed approach is twofold: on the one hand, we wish to verify that the clustering algorithm, supported by the news-modeling method and the outlier removal, is effective at separating the news stories according to their content and, thus, at providing an insightful way to

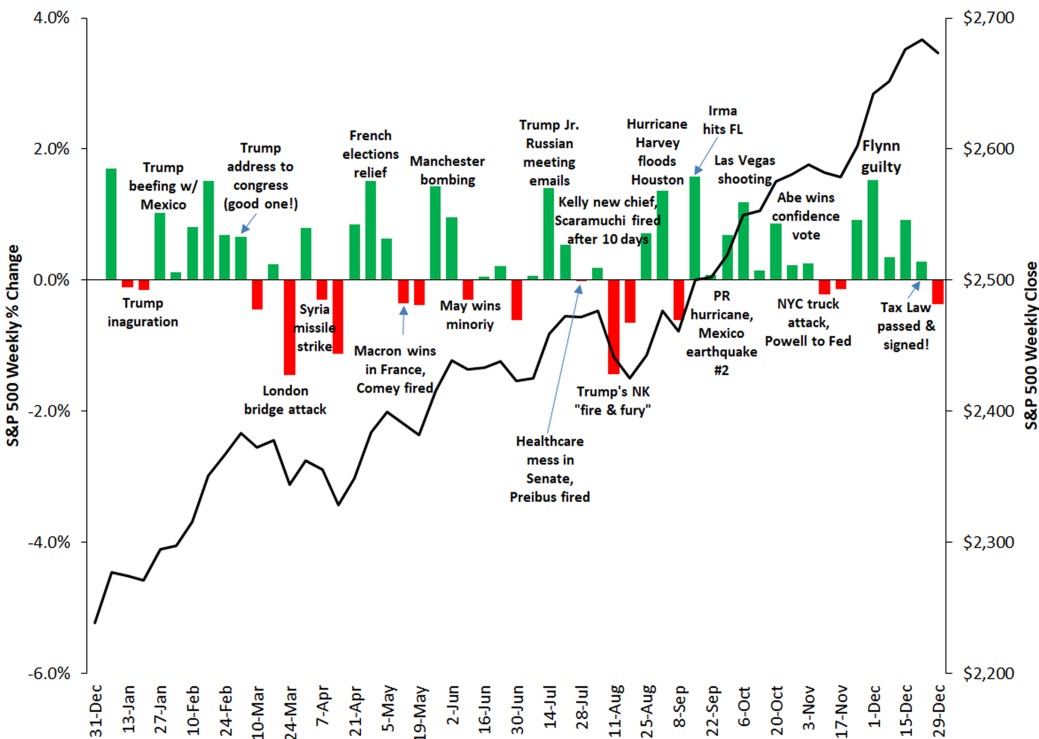

**Figure 9** Correlation between the weekly variations of the S&P 500 stock price and relevant events taking place in the U.S. and worldwide in 2017. Source: https://gordianadvisors.com/.

inspect events. On the other hand, we want to assess the accuracy of the alert-generation algorithm, in particular to confirm that there is a correlation between *hot* events spotted by our approach and remarkable real-world events. For our purposes, we performed the following set of experiments: (i) comparison of different clustering techniques; (ii) event-detection qualitative evaluation; (iii) alert-generation assessment.

Assessing the performance of an event-detection task is a daunting task, and getting a thorough quantitative evaluation is not trivial as well. This is partly due to a certain degree of subjectivity implied in the definition of an *event*, even more when we consider it with respect to a very specific context. In fact, an important occurrence such as a terrorist attack taking place in Europe, which is relevant in an absolute sense, might not be perceived as a relevant event in the sphere of U.S. finance. Moreover, to the best of our knowledge, universally recognized benchmarks of financial events are not available. For these reasons, in order to limit the subjectivity of the choice, we decided to select a list of events in a deterministic way, based on the weekly variations of the S&P 500 Index (more details on the selection method will be given in "Conclusions and Future Work"). Intuitively, we follow the assumption that important financial events are commonly associated with significant reactions of the stock markets, as suggested by the plots in Figs. 9, 10 and 11, that show the correlation between the weekly variations of S&P 500 stock price and relevant events taking place in the U.S. and in the rest of the world.

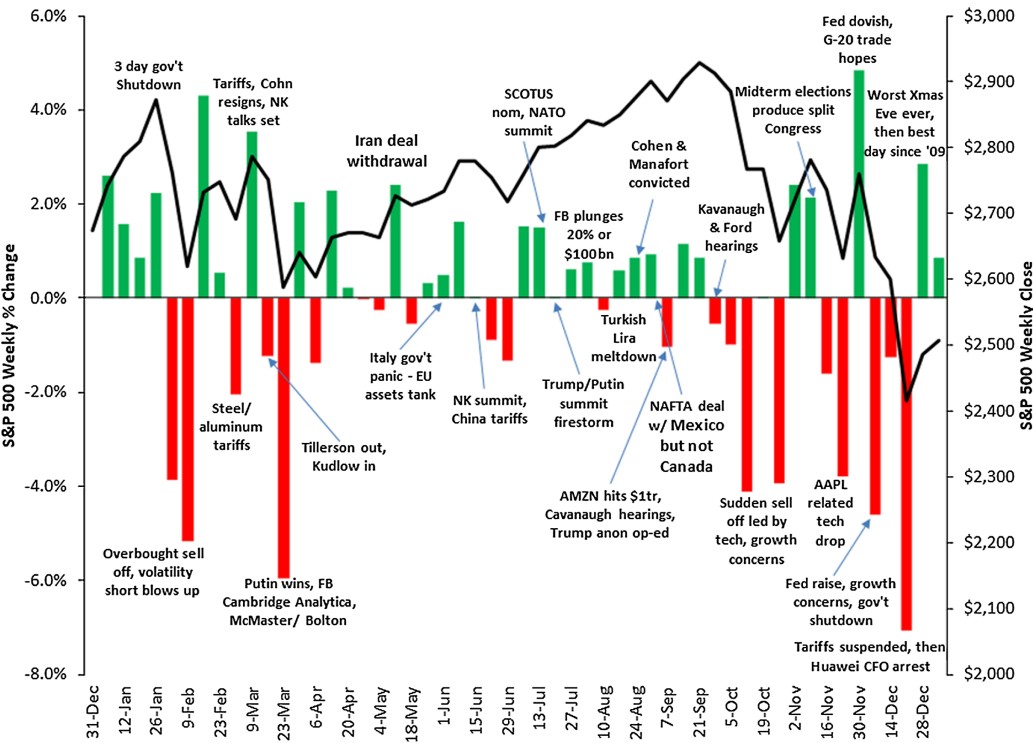

**Figure 10  Correlation between the weekly variations of the SP500 stock price and relevant events taking place in the U.S. and worldwide in 2018.** Source: https://gordianadvisors.com/.

The datasets presented in "Datasets" were filtered in order to extract the data most suited for our evaluation. Specifically, we selected from Dow Jones DNA all the news in English language published between June 2016 and March 2020, containing the keyword "Standard & Poor's" (as well as strictly related keyword like "S&P 500" or "SP500") in the title or in the body of the article. We aligned this collection of news with the Stocktwits data, which was collected in the same time-span, as already mentioned above. The whole filtered sets are thus composed of 8,403 news and 283,473 tweets. Finally, we considered the S&P 500 daily price time-series in the same years. The approach and experimental framework were developed in Python employing a set of open source Machine Learning libraries. The implementations of the agglomerative clustering algorithm, dimensionality reduction, TF-IDF method, and clustering performance metrics, were based on the popular scikit-learn library (http://scikit-learn.org); the K-Means, K-Medoids, K-Medians algorithms used for comparison were implemented through the Pyclustering library (http://pyclustering.github.io); Natural Language Toolkit (http://www.nltk.org) and gensim (http://radimrehurek.com/gensim/index.html) libraries were exploited for text pre-processing. As far as word-embeddings are concerned, we relied on the pre-computed word2vec model based on (*Mikolov et al., 2013*), trained on part of a Google News dataset composed of about 100 billion words. The model contains 300-dimensional vectors for 3 million words and phrases (http://code.google.com/archive/p/word2vec/). Throughout the experiments presented in the paper, the parameters were set as follows (if not specified

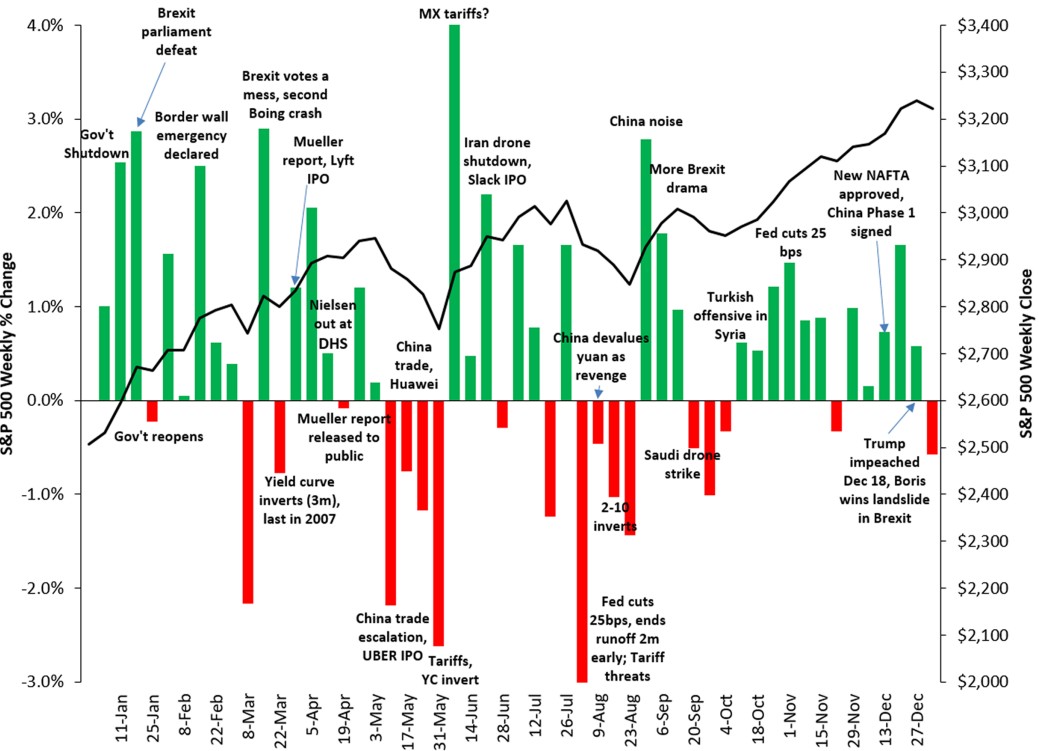

**Figure 11 Correlation between the weekly variations of the SP500 stock price and relevant events taking place in the U.S. and worldwide in 2019.** Source: https://gordianadvisors.com/.

otherwise): each daily lexicon was created on a set of news documents collected from a time window of 4 weeks, excluding all stopwords and terms that appeared in more than 90% or less than 10 documents, and the final lexicon consists of the words below the 20th and above the 80th percentiles of the ranking. The look-back window to collect the news documents to be clusterized on each day is 7 days. The *cutoff threshold* for the outlier removal stage is set to the 15th percentile. The *tweet distance threshold* for the tweet-assignment task is set to 0.5; the *alert threshold* is set to 3%. All the values of these parameters were carefully selected experimentally.

# RESULTS

In this section we will show the results we have obtained. In particular we will show the results related to the clustering algorithm, those related to three specific events, and those related to the alert-generation algorithm.

## Clustering performance evaluation

The first aspect we investigate is the choice of the clustering algorithm. As mentioned, the average Silhouette Coefficient is a standard metric to evaluate the goodness of a set of clusters. However, since it plays a role in the very construction of the clusters, we need some additional counter-checks to make the assessment more robust and less skewed. For this reason, we decided to include three more indicators in our evaluation:

- *Dunn Index*: similarly to the Silhouette Coefficient, it is a standard metric used to assess the performance of a clustering method when the ground truth is not available. It ranges from 0 to 1, with higher values indicating better clustering and is defined as:

$$Dunn\ Index = \min_{1 \leq i \leq c} \left\{ \min_{i \leq j \leq c, i \neq j} \left\{ \frac{\delta(X_i, X_j)}{\max_{i \leq k \leq c} \{\Delta(X_k)\}} \right\} \right\},$$

where $c$ is the total number of clusters, $\delta(X_i, X_j)$ is the intercluster distance between clusters $X_i$ and $X_j$ and $\delta(X_k)$ is the intracluster distance within cluster $X_k$.

- *Number of extracted clusters*: this is also a useful indicator to evaluate the quality of a set of clusters, as higher values typically suggest a better separability of the data.

- *Overlapping between the clusters' relevant words*: it is estimated by computing the Jaccard Index[2] between the lists of top-10 relevant words for each pair of clusters, and by averaging the results. A small average overlapping signifies that news documents belonging to different groups discuss different topics and, therefore, that the articles were properly split according to their semantic content.

[2] The Jaccard Index between two lists is defined as the size of their intersection divided by the size of their union

We used these metrics and the Silhouette Coefficient to compare four different techniques, namely Agglomerative clustering, K-Means, K-Medians and K-Medoids. These algorithms were executed on the same instances of the data selected for our evaluation, on each day of the time interval, using a look-back window of 1 week. Fig. 12 shows the outcome of this experiment, indicating that Agglomerative is the algorithm that leads to better performance in terms of Silhouette, Dunn Index and most remarkably in the number of extracted clusters, while the overlapping of relevant words does not differ much. Please consider that the metrics were computed only after the outlier removal phase, which is responsible for an improvement of approximately 50% of both Silhouette and Dunn Index.

### Event-detection evaluation

The results presented in the previous section, although obtained through an unsupervised validation, prove by themselves the effectiveness of the clustering algorithm at detecting events from a set of news. For illustration purposes, in this section we will carry out a qualitative analysis of the output of the clustering algorithm, focusing our attention on three specific events:

- The 2016 United States Presidential Elections (8th November 2016);
- The escalation of the U.S.-China trade war (9th May 2019);
- The outbreak of the Covid-19 pandemic (28th January 2020).

We can observe that these three events are well-known world wide and there is no need to agree on those days (we invite the reader to refer to "Proposed Approach" for an analysis of the Brexit referendum in June 2016).

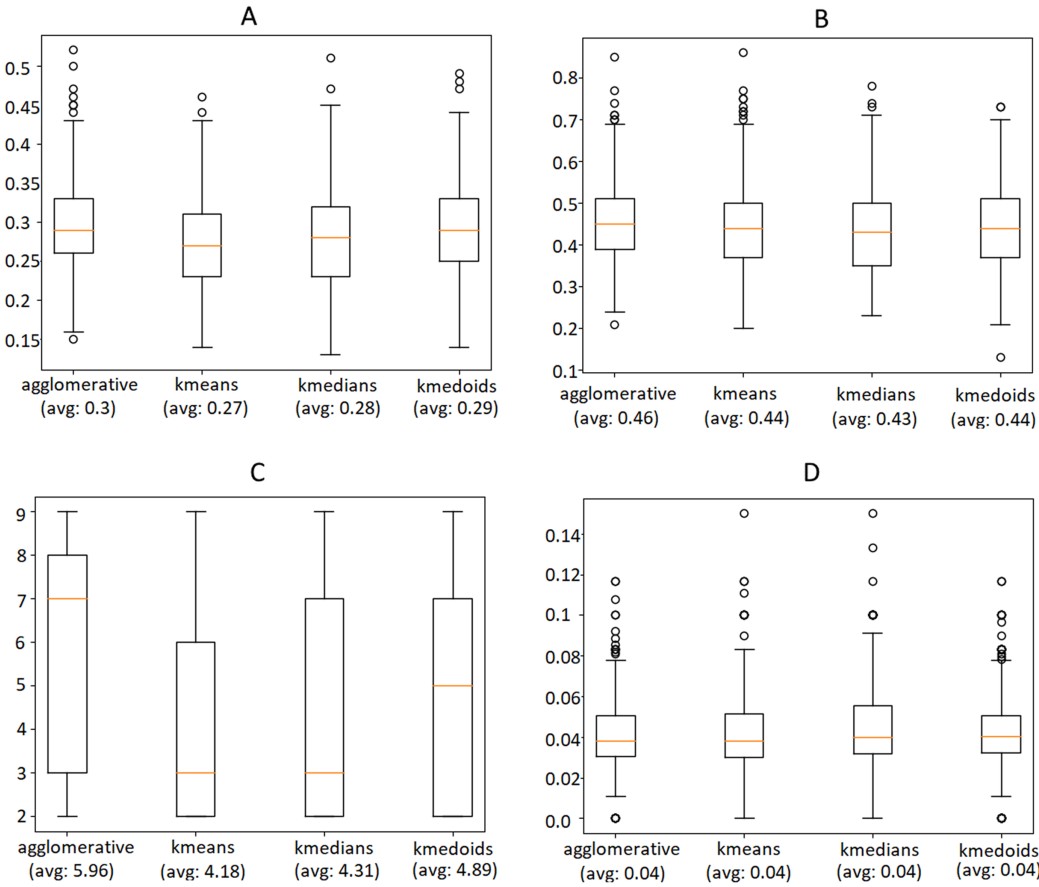

**Figure 12 Comparison of the Silhouette Coefficient (A), Dunn Index (B), number of clusters (C) obtained by different clustering algorithms and overlapping between the clusters' relevant words (D).** The horizontal orange line represents the median of the obtained scores whereas the average is indicated between parenthesis. For further details we refer the reader to the official documentation of matplotlib library: https://matplotlib.org/3.1.1/api/_as_gen/matplotlib.pyplot.boxplot.html.

For each event, we picked one date among the most significant ones: the 8th of November is the actual date of the 2016 U.S. elections, which brought to the victory of Donald Trump (http://en.wikipedia.org/wiki/2016_United_States_presidential_election); the 9th of May 2019 is a few days after Trump threatened to raise tariffs on China and just one day before U.S. actually increased tariffs from 10% to 25% (http://china-briefing. com/news/the-us-china-trade-war-a-timeline); the 28th of January is the first day on which the total number of confirmed cases of Covid-19 worldwide surpassed the one-thousand threshold, passing from 793 to 1,786 with a dramatic 125% daily change (http://covid19.who.int). We applied the event-detection algorithm on the news published in the previous week, not including that date.

In Fig. 13 we illustrate the results of the event-detection for each tested clustering algorithm. We remind that the Agglomerative clustering outperforms the others (as it can also be seen from the plots), and, therefore, we will focus our analysis on it (subfigures a, b and c).

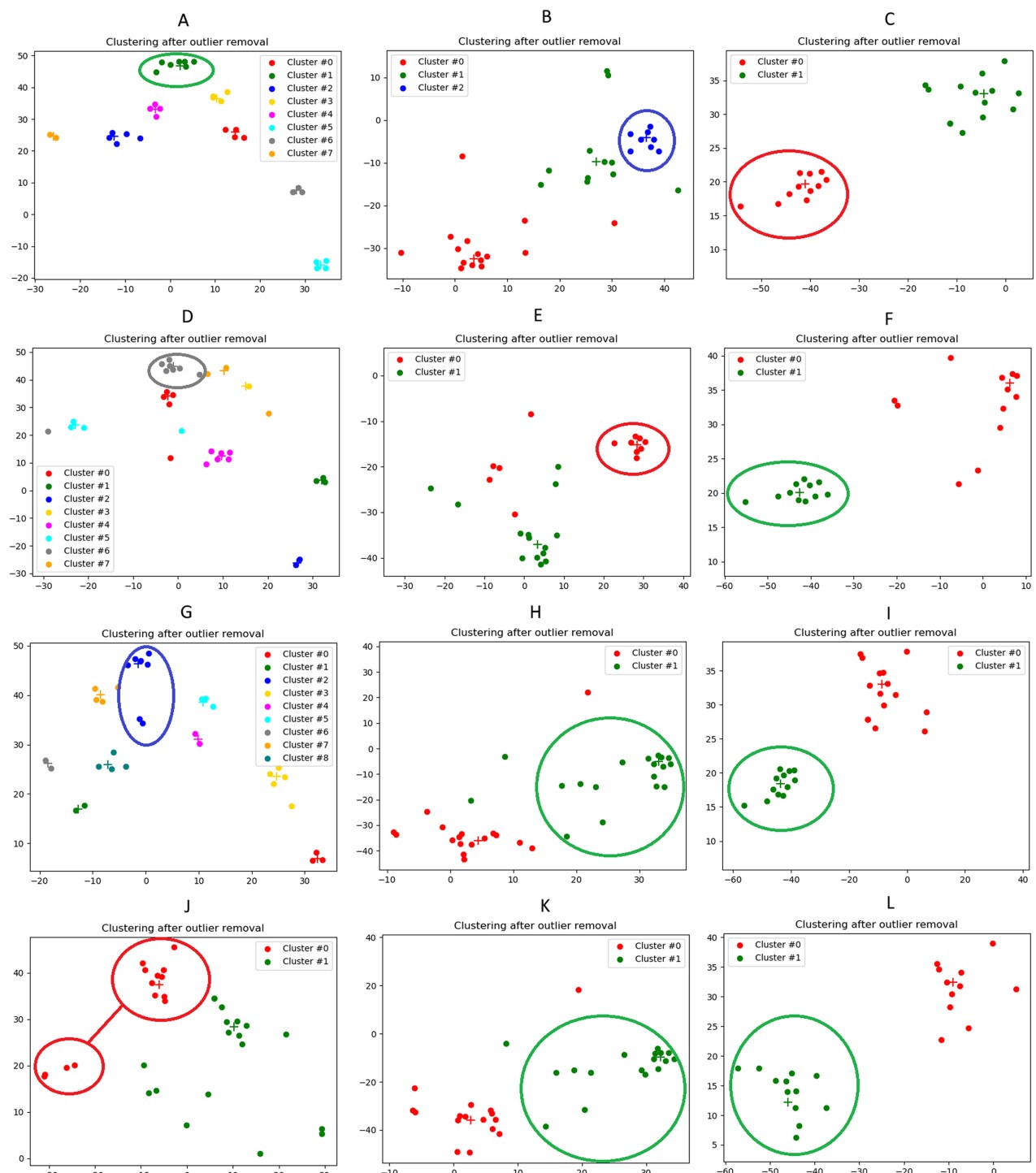

**Figure 13 Illustration of the news clusters extracted on the three case studies considered in our qualitative analysis.** (A): 2016 U.S. El. (Agglomerative). (B): U.S.-China trade war (Agglomerative). (C): Covid-19 outbreak (Agglomerative). (D): 2016 U.S. El. (K-Medoids). (E): U.S.-China trade war (K-Medoids) (F): Covid-19 outbreak (K-Medoids). (G): 2016 U.S. El. (K-Medians). (H): U.S.-China trade war (K-Medians). (I): Covid-19 outbreak (K-Medians). (J): 2016 U.S. El. (K-Means). (K): U.S.-China trade war (K-Means). (L): Covid-19 outbreak (K-Means). The cluster associated to the event is highlighted by a circle. The correspondence between cluster and event is easily understood by manually reading the relevant words and the headlines of the documents that were associated to that label. For information about the 2D visualization technique, please refer to caption in Fig. 3.

**Table 4 Lists of top-10 relevant words for the three case studies considered in our qualitative evaluation.**

**Events**

| 2016 U.S. elections | U.S.-China trade war | Covid-19 outbreak |
|---|---|---|
| clinton | tariffs | virus |
| monday | percent | covid-19 |
| trump | trump | outbreak |
| election | chinese | chinese |
| percent | united | losses |
| october | talks | impact |
| team | deal | europe |
| victory | friday | department |
| polls | imports | boeing |
| presidential | goods | hopes |

**Table 5 List of the three most relevant headlines (i.e. closest to the respective centroid) for the three events considered in the qualitative evaluation.**

| Event | Top-3 Relevant Headlines |
|---|---|
| 2016 U.S. elections | - World Stocks: Dollar, Asia Stocks Rise on Clinton News<br>- Stocks: Election Has Foreign Funds Wary—After Brexit surprise, some avoid investing in U.S. stocks until president determined<br>- Election Presents Dilemma - Markets can't price in both a Trump win and a Democrat sweep of Congress at same time |
| U.S-China trade war | - Fear of Tariffs Jolts Markets And Nerves<br>- U.S. Advisers Say China Is Reneging On Trade Accord<br>- U.S. News: Tariffs Would Hit Consumer Goods |
| Covid-19 outbreak | - World markets show signs of fever<br>- Markets on the slide as fears spread over virus<br>- Britons returning from China to be 'safely isolated' for 14 days |

From the 2D visualizations presented in Fig. 13, it can be seen that the points representing the news documents properly group up to form clusters. Interestingly enough, Fig. 13C shows a strong polarization of the news, with only two clusters obtained. This is probably ascribable to the epochal impact of the Covid-19 outbreak, that drew a considerable part of the attention of the media, with many other topics left uncovered in the press. The average Silhouette Coefficient is decidedly above 0 for all three case studies (0.28, 0.27 and 0.36, respectively), indicating a satisfactory performance of the Agglomerative algorithm. These results are confirmed by the lists of relevant words (Table 4), relevant news headlines (Table 5) and relevant tweets (Table 6), which accurately reflect the semantics of the events (these last have been generated from the agglomerative clustering output only).

**Table 6 List of the three most relevant tweets (i.e. closest to the respective centroid) for the three events considered in the qualitative evaluation.** Please keep in mind that the time interval used for the U.S. elections does not include the outcome of the polls (hence the wrong forecasts by users that initially proclaimed Hillary Clinton's victory).

| Event | Top-3 Relevant Tweets |
| --- | --- |
| 2016 U.S. elections | - Hillary Clinton Wins! |
| | - The stock market's continual favoritism of Hillary Clinton proves that she has been bought. Corruption loves company. |
| | - Markets says "Hillary Clinton Wins". Congratulation New President |
| U.S-China trade war | - Goldman Sachs think the increase in tariffs will be narrowly avoided. Odds of new tariffs at 40% if the Chinese delegation still comes. |
| | - Tariff increase on Chinese imports will take effect on May 10 - Federal Register |
| | - "Reuters: Trump's punitive tariffs will burden consumers"; yeah like it… |
| Covid-19 outbreak | - Mainland Chinese, Hong Kong stocks tumble as Covid-19 death toll rises |
| | - Second U.S. Covid-19 case is Chicago resident who traveled to Wuhan |
| | - 3M Ceo says there factories are working 24/7 making masks & Protective equipment to fight the virus. Buy your calls while there Cheap. #stocks #covid-19 |

## Alert-generation evaluation

As mentioned in Fig. 1, the Alert-generation is the last step of the proposed pipeline and is performed on top of the clustering results and the tweets assignment to the generated clusters. The accuracy of the alert-generation algorithm can be gauged in terms of how many *hot* events it is able to spot in a given ground truth. As mentioned in "Methodology and Settings", we selected the ground truth for our evaluation by looking at the weekly variations of the S&P 500 Index. More in detail, for every day $d$ between June 2016 and March 2020 we compute the variation, in absolute value, between the close price of $d$ and the close price of $d + 7$ (i.e., 7 days after $d$). This quantity is formally defined as:

$$\Delta_d = \frac{|close_{(d+7)} - close_d|}{close_d}. \tag{2}$$

The days $d$ for which $\delta_d > 0.02$ are marked as *event days*. The threshold of 0.02, which corresponds to a 2% weekly variation, is set experimentally and leads to approximately 15% of days being marked as an *event day*. Consecutive *event days* are aggregated to form events, which are thus defined as contiguous intervals, delimited by a *start date* and an *end date*. In order to avoid splitting intervals that actually refer to the same real event, we ignore interruptions of up to 3 days in the chain of consecutive *event days*. For example, if the *event days* are 2018-01-01, 2018-01-02, 2018-01-03, 2018-01-12, 2018-01-13, 2018-01-15, then the resulting events are defined by the intervals (start: 2018-01-01, end: 2018-01-03) and (start: 2018-01-12, end: 2018-01-15). We assess the recall of the alert-generation algorithm using the following method: for each (start date, end date) pair, we check if the algorithm produces at least one alert within that interval. In the positive case,

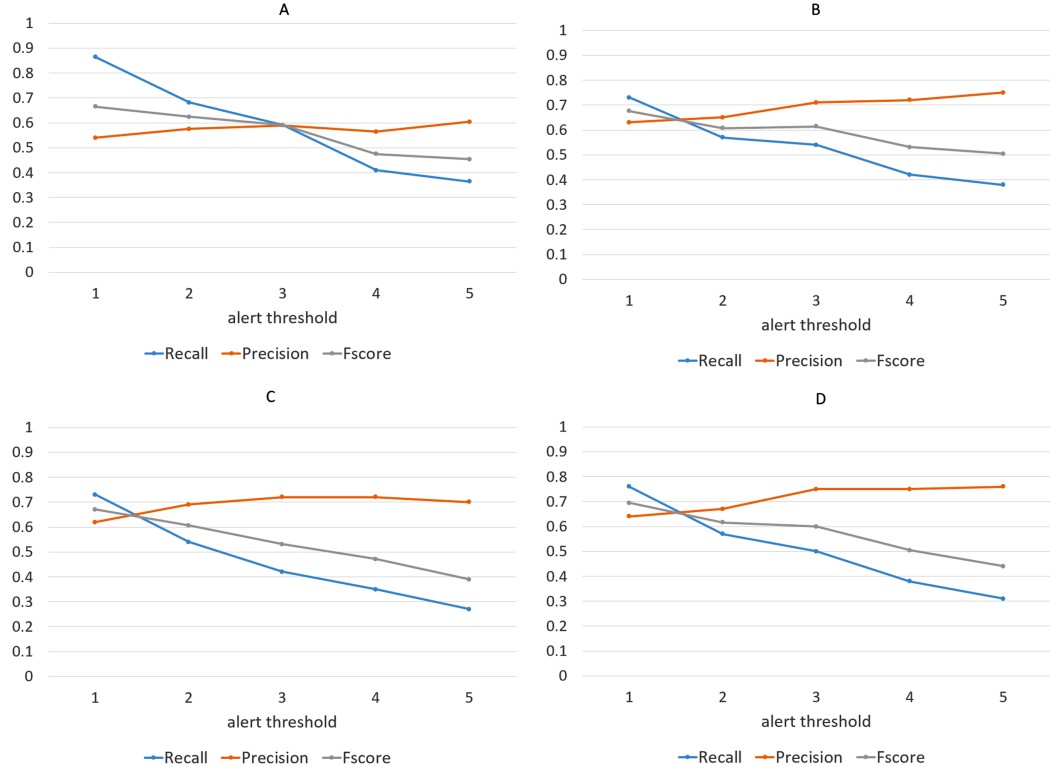

**Figure 14 Precision, recall and F-score achieved by the alert-generation algorithm for different values of alert threshold and for each of the four clustering approaches.** (Agglomerative (A), K-Means (B), K-Medians (C), and K-Medoids (D)).

the event is considered as *spotted*. The final recall is given by the number of spotted events out of the total number of events. On the other hand, to calculate the precision we iterate over the daily alerts generated by the algorithms. If an alert lies within an event interval defined by a (start date, end date) pair, then it is considered a *hit*; otherwise, it is considered a false alarm. The final precision is obtained as the number of hits out of the overall number of alerts. The F-score is computed with the standard formula: $2 \cdot precision \cdot recall precision + recall$ .

We have repeated the experiment for different values of *alert threshold*, in a range between 1 and 5, with higher values producing less alerts and thus making the algorithm more selective. Not surprisingly, recall scores become lower as the threshold is increased, while precision follows the opposite tendency, as shown in Fig. 14. Note that we have considered precision, recall and F-score for each of the four clustering algorithms, although the discussion below targets the results obtained with the Agglomerative clustering only. This is a well-known phenomenon in Machine Learning evaluation, commonly referred to as *trade-off between precision and recall*. However, it is remarkable that, with the lowest threshold, our algorithm is able to identify almost all the events listed in the ground truth, while keeping the number of false alarms relatively small (the precision is above 0.5). It is worth noting that, in this specific application field, recall can be considered more important than precision: in fact, for a trader who relies on the alert-

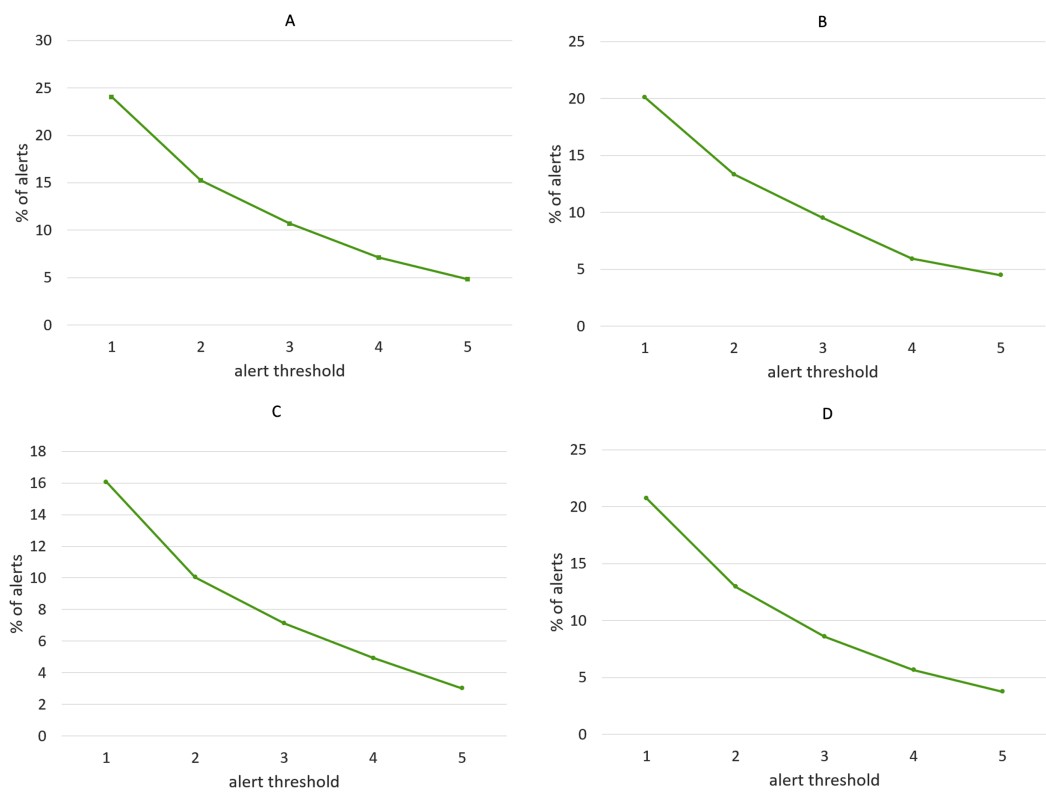

**Figure 15 Percentage of alerts produced by the alert-generation algorithm in the time-span considered for the experiments (June 2016–March 2020), for different values of alert threshold.** (A) Agglomerative. (B) K-Means. (C) K-Medians. (D) K-Medoids.

generation algorithm to make informed decisions, receiving some false alarms is arguably a lesser evil than missing relevant warnings about events that actually take place. In order to study further this phenomenon, we manually inspected several clusters that led to a false alarm, in order to understand which kinds of events cause this behavior. In many cases, we observed events like, e.g., quarterly earnings reports, that generate a big "hype" among Stocktwits users, but usually do not produce a proportional impact on the stock price. Furthermore, we calculated the percentage of days marked with an alert out of the whole period on which the algorithm was executed. Figure 15 demonstrates that, for each of the employed clustering algorithms, even with the lower thresholds, the probability of receiving an alert is still reasonably low, confirming that the algorithm is well-aimed.

An interesting finding is that, in several cases, the alert is produced with a delay of several days after the actual event took place. This can be partly ascribed to the asynchronism between newswires providers and social media (*Petrović, Osborne & Lavrenko, 2010*; *Osborne & Dredze, 2014*). In addition to this, in our specific application it is important to take into account the latency between the event itself and its effects on the market. In fact, an event might appear in financial news and on financial-related media only after some time, when its economical consequences manifest themselves. This was the case, for example, for the Covid-19 emergency: American investors, consumers and

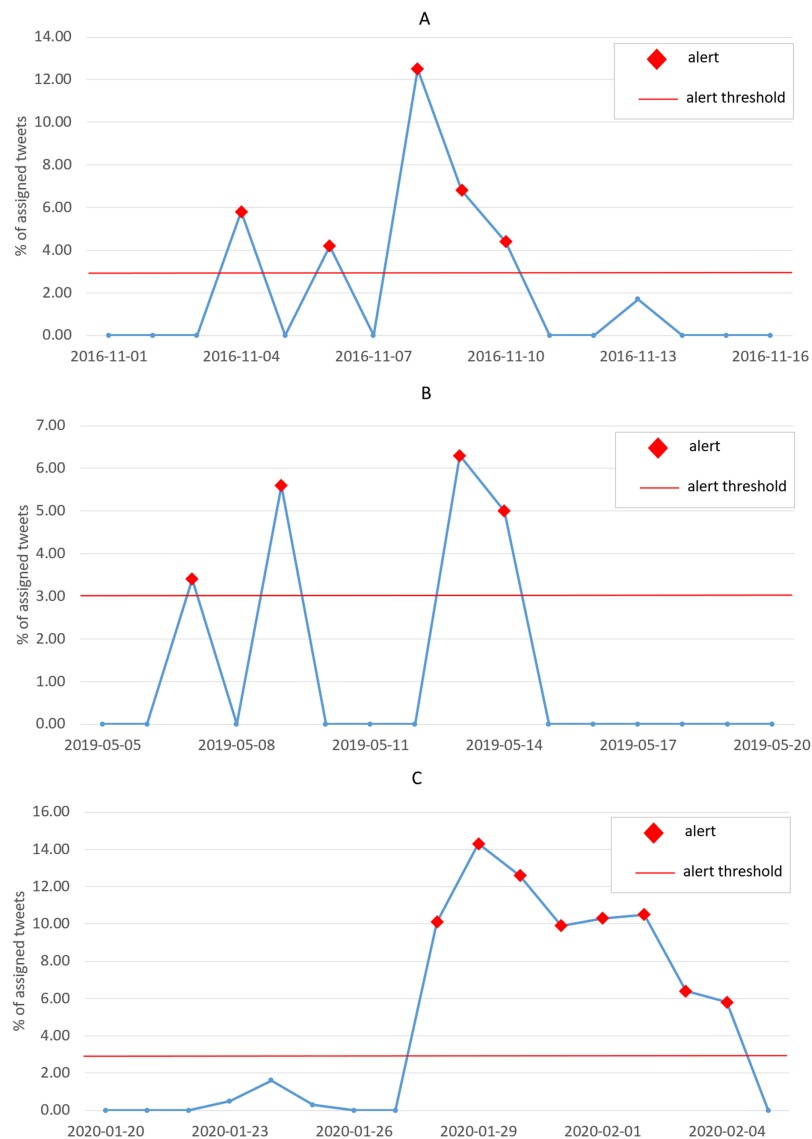

**Figure 16 Plot of the percentage of assigned tweets with respect to the overall number of published tweets, for each day in the interval around the dates of the events included in our qualitative evaluation, i.e., respectively: 2016 U.S. Presidential Elections (A), U.S.-China trade war (B), and Covid-19 outbreak (C).** The red markers indicate the generated alerts, while the red horizontal line represents the alert threshold.

market in general basically ignored news of the virus outbreak in January 2020, even though several hundred cases had already been reported and Wuhan City (first main site of the virus) had already been quarantined by the Chinese authorities. Nevertheless, starting from February the virus became the main concern of the financial-related press and media and the stock market suffered a violent plunge towards the end of the month due to COVID-19 fears (https://www.valuationresearch.com/pure-perspectives/covid-19-event-timeline/).

To conclude the visual inspection of the events discussed in the previous section, Fig. 16 illustrates the plot of the percentage of total assigned tweets, whose behaviour determines

the generation of the alerts (as explained in "Alert Generation"). It is straightforward to notice that the curves reach their peaks in correspondence of the date of the event for all three case studies, further confirming the sensitivity of our approach.

## CONCLUSIONS AND FUTURE WORK

In this work, we proposed an event-detection approach tailored for financial applications that leverages the integration of traditional newswires and social media data in order to extract information about real-world events, on a daily basis. Starting from a specialized domain-aware representation of textual data obtained through *ad-hoc* lexicons and word-embeddings, our pipeline is able to identify groups of semantically related news by means of a hierarchical clustering algorithm. An outlier-removal module refines the clusters by discarding misclassified documents, so that a noise-free, meaningful representation of events can be computed. At this point, the news clusters are enriched by data coming from social media, with the goal of estimating the impact of events on public opinions. Despite the defined tweet distance threshold that should avoid this case, it might happen that tweets containing different information are associated with the same cluster triggering the presence of a hot event that would correspond to a false positive. Although this condition has not occurred within our experiments, we will investigate it further in future works we are headed. Finally, by monitoring the activity on social media platforms, an alert-generation algorithm produces warnings to the final users, in order to make them aware of potentially relevant events that have a big resonance on the Internet. To identify the events of a day $d$, our proposed approach generates the lexicon out of news articles and stock data information of previous days up to $d-1$ without looking at the future. This makes our approach suitable for real-time event detection.

One of the advantages of the proposed approach is that, although it is domain-specific, it can be easily extended to various application fields with minimum modifications. However, in this work we described the specific pipeline and experimental framework that we implemented for the financial sphere. More in detail, word2vec models can be trained ad-hoc on text corpora in other languages, as the algorithm itself is not language-dependent. As an example, libraries such as spaCy (https://spacy.io/) provide pre-trained word-embedding models in 15 languages. Although the approach is scalable and does not have high computational times, each of its steps can be run on different machines by exploiting its pipeline architecture. Also, big data frameworks such as Apache Spark, Apache Hadoop, ElasticSearch can be leveraged and managed by cloud systems (e.g. Amazon AWS) to further make the approach faster especially if the input dataset grows exponentially.

We validated our approach by means of a qualitative and quantitative evaluation, based on Dow Jones' *Data, News and Analytics* dataset, a stream of data collected from the Stocktwits platform and the stock price time series of the S&P 500 Index. Our experiments show that the approach is effective at identifying clusters of news that correspond to relevant real-world events and at extracting meaningful properties about the associated topic. Besides, the alert-generation algorithm produces warning about *hot* events with a satisfactory accuracy, covering the majority of financial events taking place in the real

world and keeping the number of false alarms relatively small. An added value of our evaluation is given by the visual inspection of a selected number of significant real-world events, starting from the Brexit Referendum and reaching until the recent outbreak of the Covid-19 pandemic in early 2020.

One of the applications we envision is the creation of a set of financial indicators that can help improving the accuracy of existing robo-advisory and robo-trading systems. The idea is that identified hot events should be associated to high stock variations and this information might be leveraged to further tune financial forecasting systems.

In the future, we intend to carry out a deeper inspection of the temporal aspects related to the event-detection task. In particular, we want to gain a better understanding of the effect produced by parameters such as the size of the time windows used for the lexicon creation or for the clustering algorithm. Together with this, we will evaluate the timeliness of the alert-generation algorithm, with the goal of reducing the delay of the generated warnings with respect to the actual starting moment of the events. Another aspect that deserves further investigation is the method used to represent social media data in a vector space. Specifically, we intend to refine the representation of tweets by applying pre-processing techniques that are required by the specificity of the language forms commonly employed by users on Internet platforms. These methods include the assessment of the veracity and reliability of the published content and the detection of *slang*, grammatical mistakes, misspellings and abbreviations. Last but not least, we would like to take full advantage of the power and benefit that Semantic Web technologies bring: as such we would like to employ ontologies and best practices of the Semantic Web for the extraction and identification of particular events in order to improve further the obtained clustering. The employment of big data frameworks previously mentioned should address potential computational or scalability problems we might encounter.

### Funding
The Centre for Advanced Studies at the Joint Research Centre of the European Commission provided support during the development of this research work. The funders had no role in study design, data collection and analysis, decision to publish, or preparation of the manuscript.

### Grant Disclosures
The following grant information was disclosed by the authors:
The Centre for Advanced Studies at the Joint Research Centre of the European Commission.

### Competing Interests
The authors declare that they have no competing interests.

## Author Contributions

- Salvatore Carta conceived and designed the experiments, analyzed the data, performed the computation work, prepared figures and/or tables, authored or reviewed drafts of the paper, and approved the final draft.
- Sergio Consoli conceived and designed the experiments, analyzed the data, authored or reviewed drafts of the paper, and approved the final draft.
- Luca Piras conceived and designed the experiments, performed the experiments, analyzed the data, performed the computation work, prepared figures and/or tables, and approved the final draft.
- Alessandro Sebastian Podda conceived and designed the experiments, performed the experiments, analyzed the data, performed the computation work, prepared figures and/or tables, authored or reviewed drafts of the paper, and approved the final draft.
- Diego Reforgiato Recupero conceived and designed the experiments, analyzed the data, performed the computation work, authored or reviewed drafts of the paper, and approved the final draft.

## Data Availability

Code is available at GitHub:

https://github.com/Artificial-Intelligence-Big-Data-Lab/Event-Detection

Data is available at: Stocktwits:

https://stocktwits.com/

News data (commercial) is available at Factiva Dow Jones:

https://professional.dowjones.com/factiva/.

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
