# Peer review of "Event detection in finance using hierarchical clustering algorithms on news and tweets"

_PeerJ Computer Science, doi:10.7717/peerj-cs.438_

## Round 0.1 · original submission · Major Revisions

Please enhance the benchmark comparison in the results section.
Please discuss more about the applicability of the research.
Please improve the writing style.

Reviewer 1 ·

Basic reporting

In general, the manuscript reads very well. The English (wording, grammar) is spot-on and the authors have clearly made an effort to make everything as clear as possible. The article is well-structured according to the standards and seems to be well-balanced in terms of lengthiness of the separate sections. To my taste, you could consider separating Section 4 into implementation and experiments, and subsequenlty each experiment into set-up and results, just to clarify the article even further and improve readability. Figures are of good quality and provide additional clarity to the story. Minor remarks: the explanation of boxplots in Figure 12 is redundant and can be removed. Positioning of the figures and tables is also a bit awkward at times, but can be fixed in the typesetting / editing phase.

The literature review seems thorough, although I am missing some relevant work that is targeted to the actual implementation of event detection in financial algorithms, e.g.:
- A News Event-Driven Approach for the Historical Value at Risk Method: Hogenboom et al. in: Expert Systems With Applications, 42(10):4667-4675, 2015.
- An Automated Framework for Incorporating News into Stock Trading Strategies: Nuij et al. in: IEEE Transactions on Knowledge and Data Engineering, 26(4):823-835, 2014.

In fact, the authors do not make a clear distinction between "historical" and "real-time" detection of financial events. It would be nice if this is incorporated into the literature review.

The work is self contained. I would like the differences with the 2020 conference paper to be explicitly stated in this article.

Experimental design

The title and abstract seem to indicate a focus on event detection. However, the article mainly focuses on the clustering part of the problem. Alert generation (which in my view are the "actual" relevant events) is only a small, final step.

The research question remains implicit, and I would like to see it defined explicitly. The contributions, however, are very clear. The NLP-driven clustering-based approach, to my knowledge, is novel in the (financial) event detection scene. The definition of events as clusters of news items / tweets makes sense, and the fact that they are based on extracted words seems logical to me as well. The fact that the authors additionally take into account time is essential. Moreover, I like the fact that domain-specific lexicons are created ad-hoc, as most existing approaches in fact make use of pre-defined lexicons, with or without an update mechanism. Therefore, it would be nice to improve the lexicon generation description in Section 3, as it is not explained very well. For example:

"For each news article in this set, we extract the text, consisting of the title, the snippet and the full body
of the article, and then we perform some standard pre-processing techniques on it, such as stop-words
removal, stemming and tokenization. In addition, we remove from the corpus all the words that appear
too frequently and too infrequently, according to given tolerance thresholds."

These sentences are a bit puzzling. Which algorithms are used for pre-processing? Later on, in Section 4, I understand the authors are using standard Python packages, but they have to make sure they know which algorithms are implemented. As methodology and implementation are in separate sections (which I think is very nice), I would suggest the authors already include the applied algorithms in the methodology section, and simply state in Section 4 that the algorithms are implemented using the Python packages.

The alert generation is flawed, as these are only based on the number of mentions in tweets / news messages. I understand that this is an indication of the importance of the event, but large clusters can in fact be very general clusters. The number of mentions is directly related to the number of clusters. Secondly, why do the authors make use of today's number of mentions, but ignore next day's stock rate differences? They are an important feature of the clusters, so why not here? To me, it would make sense that hot events have a larger impact on stock rates than regular events.

Finally, it is worth mentioning that in general, the methodology is described rigorously, which enhances the reproducibility of the research sufficiently.

Validity of the findings

Experiments performed in the article are thorough and are based on a large timespan of data. I am happy to see that the authors not only performed a quantitative, but also a qualitative analysis. In my view, the latter is absolutely necessary. One thing that remains unclear, is how the real-world (ground-truth) events have been determined. Did they determine the events based on the output clusters? Or did they create them beforehand? Is there an inter-annotator agreement or the like?

One major thing that is lacking, is an evaluation of the performance against other state-of-the art approaches. I think this is crucial for the acceptance of this paper. Other than that, the evaluation of the methodology is very thorough and statistically sound, though.

The code is available in a GitHub repo, which is very nice. At first sight, the data seems to be available, but I have problems identifying the full content. Some folders mentioned in the readme, i.e., "lexicons" and "word2vec_data" are missing. I believe for the former folder, the generation script has to be executed, and for the latter folder, large external datasets have to be downloaded. It would be nice to create stub folders in the repo to avoid some confusion.

As the research question remains a bit vague, the conclusions are also quite general. I am also left with questions, i.e., does this easily port to other languages? Or is the approach heavily dependent on available word2vec models? How does the performance stack up against existing work? Does the proposed algorithm scale well? What kind of applications are envisioned?

Future work suggestions are valid. Especially temporal aspects are worth looking into. If by Semantic Web technologies the authors mean ontologies, I also believe they could help out. In essence, you are including predefined domain knowledge, which should balance out some of the artifacts of word2vec-like approaches. However, you have to keep in mind the performance hit you might suffer.

Additional comments

The proposed method seems to heavily rely on Python packages. Have the authors thought about scaling issues? For instance, the document-term matrices that are generated suffer from the curse of dimensionality. What is needed for gigabytes or petabytes of data? Have they considered cluster-computing, or even map-reduce? What kind of implications are there for performance?

The applicability of the research in practice is a bit questionable, as the analysis is a historic one. It is not easily changed either, because one of the main clustering features is stock variation the day *after* the tweet or news article has been published. This makes it hard to apply in a trading environment, as the focus there is on the here & now, and of course the future. As far as I am concerned, the merit of this work lies more in the (automatic) application in existing algorithms, but the only application I see now is historical analysis. It would be nice if the authors discuss this in more detail.

·

Basic reporting

The article proposes an event detection methodology which is time dependent and domain specific. The domain of application is financial markets where the events are extracted from financial news. The methodology consist of 3 main steps:
1. to create a time and domain-specific lexicon
2. clustering and filtering to distinguish relevant events discussed in the news
3. to spot hot topics by leveraging the news topics (events) with social media data (basically count the volume of talk for an event in Twitter)

The article is very well written, I found no grammatical mistakes, highly professional English is used. I must also praise the extensive and well-commented background literature (sect 2 Related Work) provided by the authors for it gives a good survey of the state-of-the-art of Event detection algorithms, and puts their work in modern perspective. Against this background literature this article is certainly a contribution to the field of Event detection.


Now, I have some suggestions to make the description of the proposed approach for event detection more rigorous and founded on statistical theory.
My main concern is on step 1 the lexicon generation for which it is not clear why the heuristic proposed for selecting terms will work in general (in a formal sense, not just in some practical scenarios as shown in the Experimental section). I do note that the lexicon generation method comes from previous work by the authors (see in their references: Carta et al (2020)) which I read through.
Neither there or in this paper the method is given a statistical support/foundation, only empirical assessment.

First a suggestion to make the description of step 1 (Sec. 3.2 .Lexicon generation) more clear and unambiguous.
line 341: "each row corresponds to a news article **and date**, and each column ..."
(The words "and date" should be added: Each news must have its time-stamp explicitly for the next step described in lines 343-346 where return is associated to the terms)
L 344-346, I advice to use other letter for the "day after the article was published" (e.g. c), since d is the date at which we want to discover hot events and for which we are looking at news published on days c \in [d-l;d-1]. In this way it becomes evident that we are looking at news and returns at different time-stamps, whereas the current formula (1) gives a first impression that we are only looking at news on day d-1 (before the events of interest)

Now I offer the authors an argument that may connect their construction to the theory of "sure screening" (Fan and Lv, 2008), hence sustaining it on formal statistics.
I see their lexicon generation equivalent to computing the frequency f(j) with which a term j co-occurs with a positive value of return.
Assume in the period [d-l;d-1] the algorithm collects N articles, out of which n<N contain the term j (articles can be sentences) . Then
f(j) = (1/n)*\sum_{1 \le k \le N} X_k(j)*\Delta_c(k)
where X_k(j) is a dummy variable for whether term j appears in article k and Delta_c(k) is the return on the day c for article k.
Thus, in this form, f(j) is the slope of a cross-article regression of
\Delta_c = (\Delta_c(1), ..., \Delta_c(N)) on the dummy variable X(j) = (X_1(j), ..., X_N(j)).
i.e. f(j) are coefficients of a marginal regression [Genovese et al, 2012].
By next sorting them by decreasing scores and selecting those whose value is over (resp. under) some threshold t+ (respect. t-) -which is similar to taking the first n and the last n-, the authors are doing "marginal screening", a form of variable selection which is proven to be more efficient than the Lasso and with good statistical accuracy. What I mean with this last sentence is that if S is the index set of true sentiment charged terms (positive and negative), and
\hat{S} = { j : f(j) \ge t+, or f(j) \le t- } then under certain conditions
Prob(\hat{S} = S) --> 1 as N and the number of terms go to infinity . This is the sure screening property (Fan and Lv).

This guarantees consistency of the procedure for screening terms for the lexicon (that almost surely are the true sentiment charged terms correlated with price variation) and if you were to work out the proof of the asymptotic result you'll get along the way an expression for the threshold, the number of articles and the terms needed.
I don't see how to prove this in this context easily (and I guess is difficult), so I would not pose this as requirement for publishing this article. However, it would be nice that the authors check this framework of sure screening and include at least an explanation like the one I am offering to mathematically sustain their lexicon construction heuristic.

References:
Fan, Jianqing, and Jinchi Lv, 2008, Sure independence screening for ultrahigh dimensional feature space, Journal of the Royal Statistical Society: Series B (Statistical Methodology) 70, 849-911.

Genovese, Christopher R, Jiashun Jin, Larry Wasserman, and Zhigang Yao, 2012, A comparison of the lasso and marginal regression, Journal of Machine Learning Research 13, 2107-2143.


Minor comments:
In the legend of Figure 3 : "The bi-dimensional visualization ... by means of tSNE, ..." What is t-SNE? (I am not familiar with this nomenclature) . I short line explaining it, or a reference where to look it up will be appreciated.

Experimental design

No cooment

Validity of the findings

No comment

Additional comments

In the Introduction (lines 85-86) you claim the approach (of event detection) "can be applied to various domains with minimum modifications". However I see that a key component is to have some form of quantifying the behavior of the domain one want to characterize the important events, in order to construct a sentimental-lexicon conditioned to that target characterising the domain. In the case of financial markets, represented by the market index (e.g SP500), this quantification of its behavior is the variation of the price of the index (or return). For another domain (e.g. politics) what could be this numeric feature that characterizes the domain and that serves as driver for the lexicon generation? An example would be a plus for this nice and interesting event detection proposal.

---

## Round 0.2 · Minor Revisions

Please further improve the writing of the manuscript specifically according to the attached annotated manuscript.

Reviewer 1 ·

Basic reporting

No comment

Experimental design

No comment

Validity of the findings

No comment

Additional comments

The revised version is a major improvement over the initial manuscript. The authors have clarified all of my questions and addressed them in the manuscript appropriately.

Two remarks:
1) The new references in sections 2.4 and 3.1 plus figure 3 seem to have been parsed incorrectly. There are question marks instead of author/year statements. This seems only to be the case in the tracked changes document, but a final check of all references might be good anyway.
2) I am not fond of the constructions "the reader notes" and "the reader notices". Perhaps the authors could consider different phrasing?

All in all, job well done!

·

Basic reporting

This is a re-review of the article that I have already revised in fully. My previous praise about the articles proposal, experimental design and results carries through to this revised version.
In this updated version of their orignal article, the authors have complied with all suggested corrections (my own remarks and other reviewer) improving as desired their original manuscript. However there are still some minor typos to be fixed. After fixing this I feel confident about publishing their work.

I will list my new comments below and for further reference I will submit the manuscript with my comments registered in it so that they can be easily found.

- The authors have included the statistic explanation of the lexicon generation process I suggested to them. Good. However, I feel it should not go in section 3.1, where they put it, but rather at the end of section 3.2. In section 3.1 a general overview of the system is given, and expanding here on the mathematics specificity of the lexicon generation breaks the generality of the description. It looks out of context, and in fact the notation needed (Delta(k), ...) has not been introduced, as it is part of sec. 3.2.

- In the same paragraph (p 11) the equation for f(j) please replaced * for \cdot. To use * for product is unusual, and also you use \cdot later on for other products.

- insert $t^+$ ($t^-$), after "specified threshold"

Now
p 5 (abstract) l 28: remove "we create". This should be "the algorithm dynamically builds a lexicon by looking ..."

l 289: although --> accordingly,

l 290: "considering the future information of a day".. substitute by "considering information beyond the day "

l 736: "The reader notices " substitute by The reader should notice

Experimental design

No comment

Validity of the findings

no comment

Additional comments

Nice work!

---

## Round 0.3 · accepted · Accept

The manuscript has been well revised. When preparing the final files, please replace all figures in ".png" fomat to ".pdf" format. PNG files are not vector illustrations, they cannot be enlarged. In contrast, PDF files are vector illustrations, they can be very high-quality. For example, when outputting the figures 6 from Origin software, please select the option, PDF, not PNG. When outputting other figures from Adobe Illustrator, please directly save as PDF files, not PNG files. In short, please save all figures in vector format.

·

Basic reporting

This is the third round of reviewing. The authors have satisfactorily made all my suggested minor corrections. Now the paper is of my entire satisfaction. Go ahead to publication.

Experimental design

Fine

Validity of the findings

Fine

Additional comments

Good job!